# Simplified synthetic routes for low cost and high photovoltaic performance *n*-type organic semiconductor acceptors

Xiaojun Li[1,2], Fei Pan[1,2], Chenkai Sun [1,2], Ming Zhang[3], Zhiwei Wang[4,5], Jiaqi Du[1,2], Jing Wang[3], Min Xiao[4,5], Lingwei Xue[1], Zhi-Guo Zhang[1], Chunfeng Zhang[4,5], Feng Liu[3] & Yongfang Li[1,2,6]

The application of polymer solar cells (PSCs) with *n*-type organic semiconductor as acceptor requires further improving powder conversion efficiency, increasing stability and decreasing cost of the related materials and devices. Here we report a simplified synthetic route for 4,4,9,9-tetrahexyl-4,9-dihydro-s-indaceno [1,2-b:5,6-b'] dithiophene by using the catalyst of amberlyst15. Based on this synthetic route and methoxy substitution, two low cost acceptors with less synthetic steps, simple post-treatment and high yield were synthesized. In addition, the methoxy substitution improves both yield and efficiency. The high efficiency of 13.46% was obtained for the devices with MO-IDIC-2F (3,9-bis(2-methylene-5 or 6-fluoro-(3-(1,1-dicyanomethylene)-indanone)-4,4,9,9-tetrahexyl-5,10-dimethoxyl-4,9-dihydro-s-indaceno [1,2-b:5,6-b'] dithiophene) as acceptor. Based on the cost analysis, the PSCs based on MO-IDIC-2F possess the great advantages of low cost and high photovoltaic performance in comparison with those PSCs reported in literatures. Therefore, MO-IDIC-2F will be a promising low cost acceptor for commercial application of PSCs.

[1] Beijing National Laboratory for Molecular Sciences, CAS Key Laboratory of Organic Solids, Institute of Chemistry, Chinese Academy of Sciences, 100190 Beijing, China. [2] School of Chemical Science, University of Chinese Academy of Sciences, 100049 Beijing, China. [3] Department of Physics and Astronomy and Collaborative Innovation Center of IFSA (CICIFSA), Shanghai Jiaotong University, 200240 Shanghai, China. [4] National Laboratory of Solid State Microstructures, School of Physics, and Collaborative Innovation Center of Advanced Microstructures, Nanjing University, 210093 Nanjing, China. [5] Synergetic Innovation Center in Quantum Information and Quantum Physics, University of Science and Technology of China, 230026 Hefei, Anhui, China. [6] Laboratory of Advanced Optoelectronic Materials, College of Chemistry, Chemical Engineering and Materials Science, Soochow University, 215123 Suzhou, Jiangsu, China. These authors contributed equally: Xiaojun Li, Fei Pan, Chenkai Sun. Correspondence and requests for materials should be addressed to C.Z. (email: cfzhang@nju.edu.cn) or to F.L. (email: iamfengliu@126.com) or to Y.L. (email: liyf@iccas.ac.cn)

Solar cell, which transforms the inexhaustible solar energy into electricity, is one of the most promising clean and renewable energy sources. Currently, the commercialized Si-based solar cells can achieve high efficiency but are produced through complicated energy-consuming processes with serious environmental pollutions during the preparation and purification of silicon crystals and production of the solar cells[1–3]. In comparison, the third generation organic solar cells (OSCs) possess the advantages of simple device structure, low-cost solution processing and capability to be fabricated into flexible and semitransparent devices[4–7]. Currently, considerable progress in the design and synthesis of high performance photovoltaic materials and optimization on the device structure has led to rapid increase of the power conversion efficiency (PCE) of OSCs[8–16]. Especially, the narrow bandgap $n$-type organic semiconductor ($n$-OS) small molecule acceptors[17–25], have demonstrated excellent photovoltaic performance in combination with medium or wide bandgap $p$-type conjugated polymer as donors[26–31]. PCE of the polymer solar cells (PSCs) with the $n$-OS as acceptor has boosted to over 13%[32–34], which reached the threshold for application of the PSCs. However, at present, most of the high performance donor and acceptor photovoltaic materials have complicated molecular structures[23,35,36]. And their synthesis is confronted with verbose synthetic processes, intractable conditions, multiple purifications and low yields, which results in high-energy consumption and high cost.

Actually, nowadays the fabrication of large area PSCs still relies mainly on poly(3-hexylthiophene) (P3HT) as donor[37], because P3HT is readily synthesized in large scale with controllable molecular weight and low cost. However, the photovoltaic performance of P3HT is limited to the PCE of <8% due to its relatively high-lying HOMO energy level and narrow absorption band[38,39]. Therefore, it is of crucial importance to simplify the synthetic processes and reduce the cost of the high performance donor and acceptor materials for the application of PSCs.

In the reduction of synthesis cost of the high performance photovoltaic materials, one way is the design and synthesis of the photovoltaic materials with simple structure like P3HT. For example, recently, our group reported a low cost conjugated polymer donor PTQ10 with simple D-A structure, two-steps synthesis and high overall yield of 87.4%[40]. Another way to reduce the cost is to simplify and optimize the synthesis process of the reported high performance photovoltaic materials, such as by designing simplified synthetic route, selecting appropriate catalyst, simplifying purification process, etc.

Nowadays, the most representative and widely used high performance $n$-OS acceptors are ITIC[41] and IDIC[42]. IDIC with the alkyl side chains on IDT core possesses the advantage of the smaller fused-ring in its central unit in comparison with ITIC, which makes IDIC have the potential to be the low cost acceptor. However, the reported synthesis method of IDIC needs complicated multi-steps with cumbersome post-processing[43], such as Friedel–Crafts acylation reaction and Wolff–Kishner reduction, which result in low yield and high cost. Meanwhile, the side reaction of dehydration on the alkyl chain[44], which is different from the ring-closure reaction of aromatic groups, may result in low yield of its central fused ring unit in the Friedel–Crafts alkylation ring-closure reaction in the synthetic route of IDIC (Supplementary Fig. 1). Thus, it is necessary to provide a better method for the synthesis of the alkyl side chain $n$-OS acceptors.

Here, we report a synthetic route to simplify and optimize the synthetic process of the central fused ring unit of IDIC by using the catalyst of amberlyst15, which could reduce the cost of IDIC. Then, we increased the yield of the central fused ring unit by introducing alkoxy substituents on it and synthesized a fused ring unit MO-IDT (see Fig. 1c). We further synthesized two low cost $n$-OS acceptors MO-IDIC and MO-IDIC-2F based on the central unit of MO-IDT, and investigated their physicochemical and photovoltaic properties in detail. The PSCs based on PTQ10 as donor and MO-IDIC-2F as acceptor demonstrated a high PCE of 13.46%. The results indicate that MO-IDIC-2F is a promising low cost $n$-OS acceptor for future application of PSCs.

## Results

**Optimization of synthetic route.** The reported synthetic routes of the central fused ring unit 2 (4,9-dihydro-4,4,9,9-tetra-hexadecyl-s-indaceno[1,2-b:5,6-b′]-dithiophene) of IDIC include the following steps: coupling reaction, ester hydrolysis reaction, Friedel–Crafts acylation reaction, Wolff–Kishner reduction and nucleophilic reaction, etc. (Fig. 1a)[43,45,46]. In this process, the catalysts (such as AlCl$_3$) used in Friedel–Crafts reaction may result in side reaction of reactants, such as the break of ether bond. Meanwhile, Wolff–Kishner reduction is not suitable for base-sensitive substrates and can be hampered by steric hindrance surrounding the carbonyl group under certain conditions[47,48]. All of these hinder the selection of substrates, and the synthetic progress for the $n$-OS acceptors with alkyl side chains has been rather limited. In addition, the Wolff–Kishner reduction reaction requires a harsh reaction condition (heated at 180 °C for 24 h), which will result in high-energy consumption. Moreover, under these verbose synthetic processes, the total yield is relatively low, the yield of Compound 2 from Compound 1 (2, 5-dithien-2-ylterephthalic acid diethyl ester) is only about 31% (Fig. 1a)[43,49].

In order to decrease the cost of IDIC and optimize the synthetic routes of $n$-OS acceptors with alkyl side chains, we firstly tried to optimize the synthetic route of Compound 2 (Fig. 1). Initially we used the synthetic path A (Supplementary Fig. 2) which is similar to the synthetic route of the fused ring core of ITIC reported in literatures[50,51]. However, a relatively low yield of ca. 50% was obtained in the Grignard reaction. Surprisingly, we found that the Grignard reaction in path B (Supplementary Fig. 2) produced fairly high yield of 85% for the key intermediate compound 4. In the subsequent cyclization reaction, as we know, there are no reports on acid-catalyzed alkylated cyclization in preparing $n$-OS acceptors with alkyl side chains[42,45,46,49]. Hsu and Yang et al. performed a series of ring closure reactions by using borontrifluoride dietherate (BF$_3$·OEt$_2$)[52–54]. However, the same reaction conditions applied to the key intermediate compound mainly afforded alkene by simple dehydration (Supplementary Fig. 2). Changing the reaction conditions, such as variation of the number of equivalents of BF$_3$·OEt$_2$ and the reaction time, still resulted in these alkene molecules mainly (see Supplementary Table 1). In order to realize the cyclization reaction, we tried other acid catalysts like H$_2$SO$_4$, TsOH, BBr$_3$, AlCl$_3$, and so on. However, these efforts did not lead to any noticeable improvement and cannot effectively get target product (see Supplementary Table 1). Finally, we used amberlyst15 as the catalyst for the ring-closure reaction, and obtained the target Compound 2 in high yield of 63% (Fig. 1b). The high selectivity of amberlyst15 for the ring-closure reaction may be attributed to the high H$^+$ ion exchange capacity and large surface area of this catalyst. Under these straightforward synthetic processes, the total yield of Compound 2 from Compound 3 is increased from ca. 31% to 49% (Fig. 1). Thus, this synthetic route is beneficial to increase the yield and reduce the cost of preparation of IDIC. In addition, the catalyst of amberlyst15 can be recycled and re-used for the low cost synthesis of the acceptors.

It is well known that the existence of electron-donating group is conducive to the Friedel–Crafts acylation reaction[55]. In order to further improve the yield of cyclization reaction, we introduce two methoxy substituents on the central benzene ring of

Fig. 1 Synthetic routes of the central fused ring units. a Synthetic route of the central fused ring core (Compound 2) of IDIC, b optimized synthetic route of Compound 2 and c synthetic route of the central fused ring core MO-IDT

Compound 3′ (see Fig. 1c), and the methoxy substituents can provide orientation effect on the following cyclization reaction. The yields of the following synthesis processes were improved to 88% from Compound 3′ to 4′ and 92% in the cyclization reaction from Compound 4′ to **MO-IDT** (see Fig. 1c), which should be benefitted from the alkoxy substituents.

The detailed synthetic procedures of **MO-IDT** are described in the section Methods. Compound 5 (Fig. 1c) was synthesized according to the method in the literature[56]. Intermediate 3′ was prepared from the reaction of 5 and 6 by Suzuki reaction using Pd (OAc)$_2$ and tri-tert-butylphosphine tetrafluoroborate as catalyst at room temperature. The crude product 3′ was purified directly by washing with petroleum ether. Then the purified Compound 3′ was dissolved in THF and reacted with hexylmagnesium bromide to give intermediate 4′. The crude product 4′ without further purification was dissolved in toluene, and Friedel–Crafts reaction was selectively performed using amberlyst15 as catalyst and gave product MO-IDT in high yield of 74%, which decreases hazardous waste production. Only four steps in the synthesis of MO-IDT are simpler than that in preparing the core of IDIC with six steps (Fig. 1a)[42,44], which will greatly decrease the cost of the fused ring core, because the material cost linearly increases with the number of synthetic steps[57]. In addition, our reaction route has the following advantages: (1) coupling reactions can be performed at room temperature in 1 h and product 3′ was obtained without column separation, which reduces the difficulty of preparing materials and makes them more convenient to get; (2) in the ring-closure reaction, selection of amberlyst15 as catalyst effectively avoids the intramolecular dehydration to olefins, which is convenient for preparing the fused ring structure with alkyl side chains; (3) this process also avoided the

Wolff–Kishner reaction with harsh conditions like high temperature. Therefore, compared with the reported synthesizing process of Compound 2 (IDIC core) with high temperature, long reaction time and the complex separation process, the synthesis of MO-IDT is more efficient and convenient, so as to realize low cost. The structure and purity of the compounds were measured and confirmed by [1]H NMR and [13]C NMR spectra, as shown in Supplementary Figures 27– 36.

**Synthesis of MO-IDIC and MO-IDIC-2F.** Based on the MO-IDT core, we synthesized two n-OS acceptors of MO-IDIC and MO-IDIC-2F with the synthetic routes as shown in Fig. 2a. Intermediate 7 was prepared by Vilsmeie–Haack reaction of MO-IDT with POCl$_3$ and DMF (Fig. 2a). Subsequent Knoevenagel condensation between 7 and Compounds 8 or 9 afforded MO-IDIC and MO-IDIC-2F in high yields. Obviously, compared with the preparation of IDIC[42], the synthesis of MO-IDIC and MO-IDIC-2F possesses the advantages of less synthetic steps, simple post-treatment, and high yield, which can reduce the synthesis cost of the acceptor for the application in PSCs. The chemical structures of the two n-OS acceptors were characterized by [1]H and [13]C NMR (Supplementary Figures 37–42)[19], F NMR spectrum (Supplementary Fig. 43 for MO-IDIC-2F) and mass spectroscopy. Moreover, the single crystals of these two molecules further clarify the structures (Supplementary Figures 44 and 45). In addition, MO-IDIC and MO-IDIC-2F exhibit good solubility in common organic solvents. And the two acceptors possess good thermal stability up to 340 and 337 °C, respectively, with 5% weight loss under nitrogen atmosphere, as measured by thermogravimetric analysis (TGA, Supplementary Fig. 3), which is

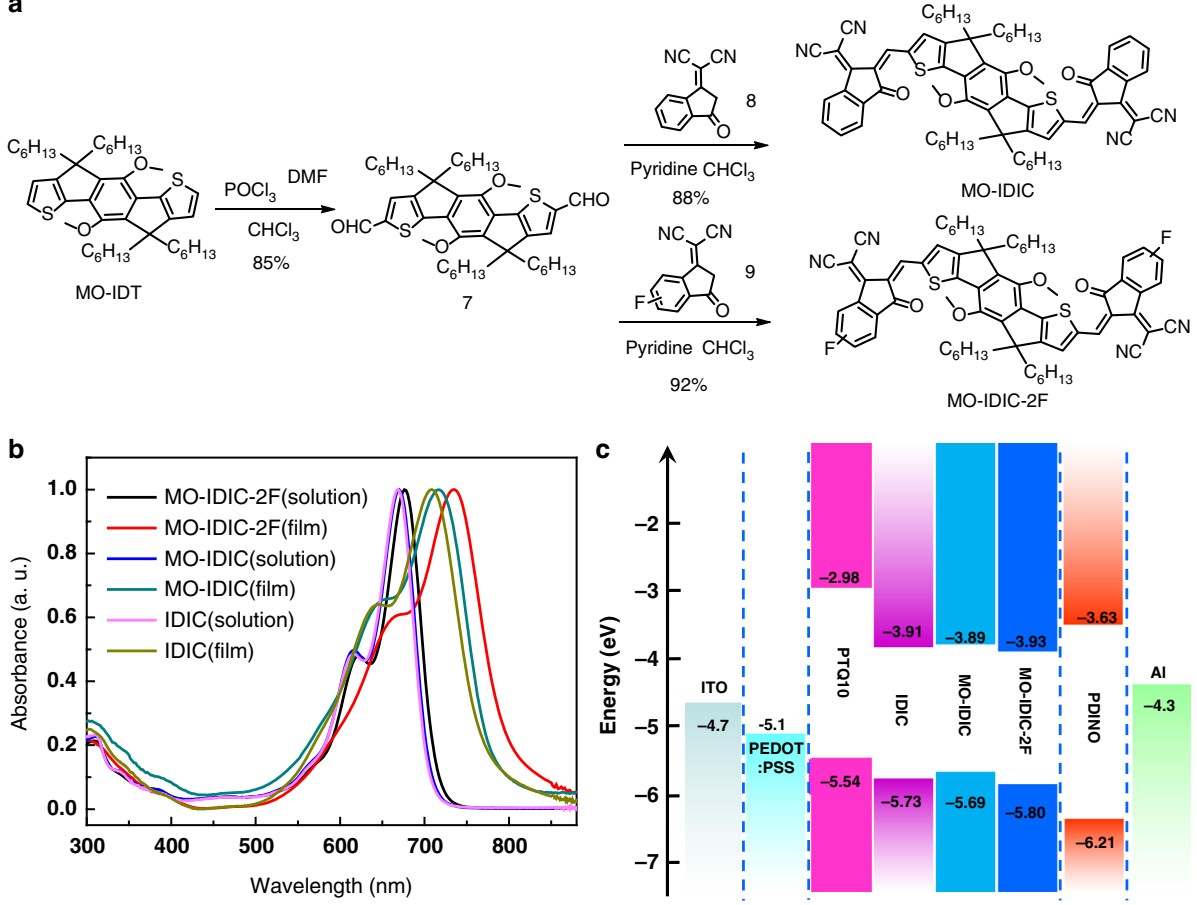

**Fig. 2** Synthetic routes and physicochemical properties of the acceptors. **a** Synthetic routes of MO-IDIC and MO-IDIC-2F, **b** absorption spectra of IDIC, MO-IDIC, and MO-IDIC-2F in chloroform solutions and films, **c** energy level diagram of the related materials used in the PSCs (the energy levels were measured by electrochemical cyclic voltammetry)

good enough for the application in PSCs from the thermal stability point of view. It should be mentioned that the stability of the two acceptors in PSC devices should be similar with that of IDIC.

**Absorption spectra and electronic energy levels**. Figure 2b shows the absorption spectra of IDIC, MO-IDIC, and MO-IDIC-2F in chloroform solutions and in thin films. These n-OS molecules exhibit strong optical absorption from 600 to 700 nm in the solutions with a maximum extinction coefficient of $2.8 \times 10^5$ $M^{-1}$ $cm^{-1}$ at 670 nm for MO-IDIC and $2.9 \times 10^5$ $M^{-1}$ $cm^{-1}$ at 677 nm for MO-IDIC-2F. In comparison with the absorption of the solutions, absorption spectra of the n-OS acceptor films exhibit a significant red-shift. In comparison with IDIC, the absorption spectrum of MO-IDIC is red-shifted slightly (red-shifted by ca. 10 nm for MO-IDIC film), probably due to the electron-donating effect of the methoxy substituents in MO-IDIC[58]. Furthermore, the absorption spectrum of MO-IDIC-2F film is further red-shifted by ca. 20 nm than that of MO-IDIC film, which could be attributed to the stronger π–π stacking interaction and more ordered aggregation in the MO-IDIC-2F film due to the strong hydrogen-bonding of fluorine atoms in MO-IDIC-2F[59]. The broader absorption spectra of MO-IDIC-2F in long wavelength range will further improve their ability to harvest solar light. The optical band gap of MO-IDIC-2F film (calculated from its absorption edge) is 1.55 eV which is lower than that (1.60 eV) of MO-IDIC.

Electronic energy levels of these two molecules were measured by electrochemical cyclic voltammetry (Supplementary Fig. 4). The $E_{HOMO}$ and $E_{LUMO}$ of MO-IDIC and MO-IDIC-2F were calculated to be −5.69/−5.80 and −3.89/−3.93 eV from onset oxidation and reduction potentials (versus Ag/AgCl), respectively (Fig. 2c). The methoxy substituents in MO-IDIC result in a slightly up-shifted HOMO (−5.69 eV) and LUMO (−3.89 eV) energy levels for MO-IDIC than that of IDIC (−5.73/−3.91 eV), which could be ascribed to the electron-rich nature of alkoxy groups of MO-IDIC. Furthermore, the fluorinated terminal units of MO-IDIC-2F lead to a slightly down-shifted HOMO and LUMO energy levels than that of MO-IDIC (Supplementary Table 2).

The electron mobilities of MO-IDIC and MO-IDIC-2F were measured by the space charge limited current (SCLC) method in electron-only devices with a structure of ITO/ZnO/active layer/PDINO/Al. The calculated electron mobilities of MO-IDIC and MO-IDIC-2F are $8.15 \times 10^{-4}$ and $1.01 \times 10^{-3}$ $cm^2$ $V^{-1}$ $s^{-1}$, respectively (Supplementary Table 3). The relatively higher electron mobilities of these two n-OS small molecules than that of IDIC[48] ($5.63 \times 10^{-4}$ $cm^2$ $V^{-1}$ $s^{-1}$) would be more conducive to the charge carrier transport in the active layer. Thus, better device performance can be expected for the new acceptors.

**Photovoltaic performance**. In order to investigate the photovoltaic properties and potential application of MO-IDIC and MO-IDIC-2F in the PSCs, we prepared the photovoltaic devices using the medium bandgap conjugated polymer PTQ10 as donor

and MO-IDIC or MO-IDIC-2F as acceptor with the device structure of ITO/PEDOT:PSS/PTQ10:acceptors/PDINO/Al. It should be mentioned that PTQ10 possesses a simpler molecular structure and relatively lower synthetic cost[40], which would benefit for the low cost preparation of the PSCs. Device fabrication details are described in the Methods. The donor/acceptor (D/A) weight ratio in the active layer of the devices and the thermal annealing temperatures were optimized, and the optimized conditions are D/A weight ratio of 1:1 and thermal annealing at 120 °C (PTQ10/MO-IDIC-based PSC) or 110 °C (PTQ10/MO-IDIC-2F-based PSC) for 5 min (see Supplementary Tables 4 and 5). And the blend active layers were spin-coated from chloroform solution under ambient condition without the use of additives.

Figure 3a shows the current density–voltage (J–V) curves of the PSCs with the optimized donor:acceptor weight ratio of 1:1, and Table 1 lists the photovoltaic performance parameters of the PSCs for a clear comparison. The PSC based on PTQ10/MO-IDIC (1:1, w/w) without thermal annealing showed PCE of 10.76% with a $V_{oc}$ of 0.975 V, $J_{sc}$ of 16.68 mA cm$^{-2}$ and FF of 66.1%. After thermal-annealing at 120 °C for 5 min, PCE of the devices was improved to 11.16% with slightly increased $J_{sc}$ and FF but a little decrease of $V_{oc}$. The device based on MO-IDIC-2F without any device post-treatment shows higher $J_{sc}$ of 18.31 mA cm$^{-2}$ and FF of 75.2% but relatively lower $V_{oc}$ of 0.899 V than that of the device based on MO-IDIC, giving a higher PCE vaule of 12.39% which is the highest efficiency for the as-cast PSCs reported so far. For further improving the efficiency of the device, thermal annealing treatment at 110 °C for 5 min was carried out, and PCE of the MO-IDIC-2F-based PSCs was improved to 13.46% with a $V_{oc}$ of 0.906 V, $J_{sc}$ of 19.87 mA cm$^{-2}$, and FF of 74.8%. In addition, the devices based on PTQ10/MO-IDIC-2F showed the higher PCE of 13.46% than that (12.09%) of the PSC based on PTQ10/IDIC-2F (see Supplementary Table 6 and Fig. 5), which indicates that the methoxyl substitution on the central core of MO-IDIC-2F is beneficial to improving its photovoltaic performance.

Figure 3b shows the input photon to converted current efficiency (IPCE) spectra of the corresponding PSCs. The high and broad photo-response over the wavelength range from 300 to 800 nm suggests that both PTQ10 donor and the acceptors made

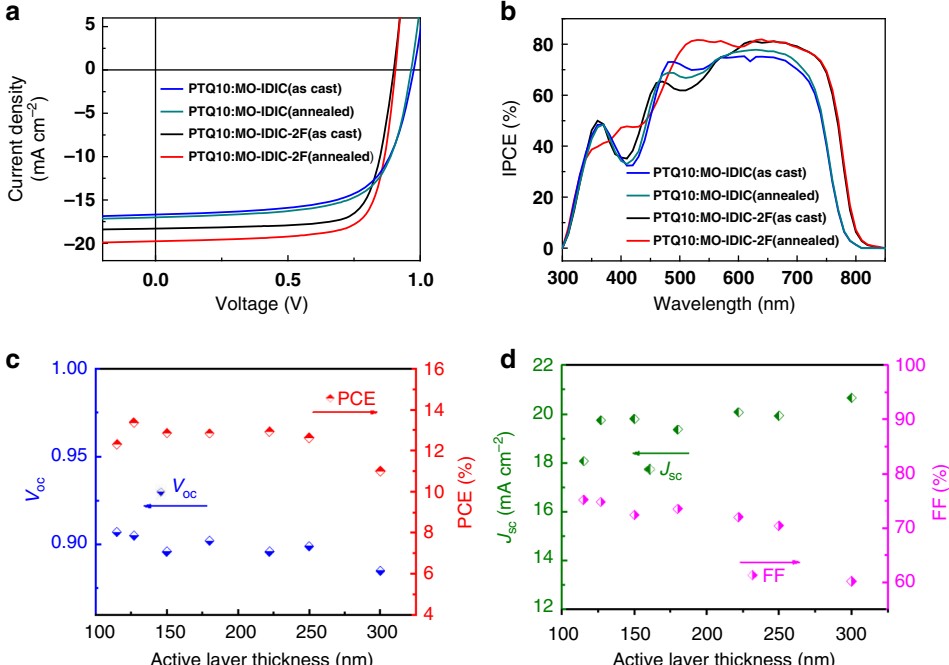

**Fig. 3** Photovoltaic performance of the PSCs. **a** J–V curves of the optimized PSCs based on PTQ10: acceptors (1:1) without (as-cast) and with thermal annealing at 120 °C (for the MO-IDIC-based devices) or 110 °C (for the MO-IDIC-2F-based devices) for 5 min, under the illumination of AM 1.5 G, 100 mW cm$^{-2}$; **b** IPCE spectra of the corresponding PSCs. Plots of **c** $V_{oc}$ or PCE and **d** $J_{sc}$ or FF vs. the active layer thickness ranging from 115 to 300 nm for the PSCs based on PTQ10: MO-IDIC-2F (1:1, w/w) with thermal annealing at 110 °C for 5 min

**Table 1 Photovoltaic performance parameters of the PSCs based on PTQ10:acceptors (1:1, w/w) under the illumination of AM1.5 G, 100 mW cm$^{-2}$**

| Acceptor | $V_{oc}$ (V) | $J_{sc}$ (mA cm$^{-2}$) | FF (%) | PCE (%) | $J_{sc}$ from IPCE (mA cm$^{-2}$) |
|---|---|---|---|---|---|
| MO-IDIC[a] | 0.975 (0.974 ± 0.004)[b] | 16.68 (15.91 ± 0.67) | 66.1 (66.8 ± 1.4) | 10.76[c] (10.49 ± 0.21) | 16.18 |
| MO-IDIC[d] | 0.969 (0.976 ± 0.006) | 16.92 (16.35 ± 0.37) | 68.1 (68.5 ± 0.6) | 11.16 (10.95 ± 0.10) | 16.38 |
| MO-IDIC-2F[a] | 0.899 (0.903 ± 0.003) | 18.31 (17.9 ± 0.56) | 75.2 (73.4 ± 1.8) | 12.39 (12.13 ± 0.12) | 17.84 |
| MO-IDIC-2F[e] | 0.906 (0.896 ± 0.005) | 19.87 (19.85 ± 0.46) | 74.8 (73.6 ± 1.5) | 13.46 (13.10 ± 0.16) | 19.12 |

[a] Without thermal annealing
[b] Average values and standard deviation data are calculated from more than 20 devices
[c] Data are the maximum values of the photovoltaic performance of the PSCs
[d] With thermal annealing at 120 °C for 5 min
[e] With thermal annealing at 110 °C for 5 min

a considerable and complementary contribution to the $J_{sc}$. The IPCE values for the MO-IDIC-2F-based devices are much higher than that of the MO-IDIC-based devices, which could be ascribed to the broader absorption in the long-wavelength range, higher electron mobility of MO-IDIC-2F and the suppressed germinate recombination in the PTQ10/MO-IDIC-2F-based device from transient absorption results (see Supplementary Discussion). The integrated photocurrent values from the IPCE spectra agree well with the $J_{sc}$ values from the J–V curves within 4% mismatch (Table 1), indicating the high reliability of the photovoltaic performance results.

Developing photovoltaic materials that tolerate thickness variations of the active layer is critical to enable large-scale manufacturing of PSCs for future application. Therefore, it is important to develop high performance and thickness-insensitive photovoltaic materials. Herein, we investigated the active layer thickness dependence of the photovoltaic performance of the PSCs based on PTQ10: MO-IDIC-2F (1:1, w/w) by changing the active layer thickness from 115 to 300 nm. Figure 3c, d show the plots of photovoltaic performance versus thickness, and Supplementary Table 7 lists the photovoltaic parameters of the PSCs. In Fig. 3c, the $V_{oc}$ values show a slight decrease with the increase of active layer thickness. Meanwhile, $J_{sc}$ values increase with increasing the active layer thickness owing to the enhanced light harvest of the thicker active layer, while FF values decrease with increasing the active layer thickness due to the increased charge recombination and series resistance of the devices with thicker active layer (Fig. 3d). The highest PCE of 13.46% is obtained from the PSC with the active layer thickness of ca.130 nm. It should be mentioned that the PCE of the PSCs with active layer thickness of 250 and 300 nm still reached the high values of 12.63% and 11.01%, respectively. The thickness-insensitivity of the photovoltaic performance indicates that the PSCs based on PTQ10: MO-IDIC-2F are suitable for large area fabrication and future application of the PSCs.

**Morphology analysis**. Grazing-incidence wide-angle X-ray scattering (GIWAXS) was used to investigate the molecular packing in neat and blend films. The GIWAXS results of neat film were shown in Supplementary Fig. 8, in which PQT10 showed a dominant face-on orientation, with a sharp (100) reflection in the in-plane direction at 0.27 A$^{-1}$ and π–π stacking in the out-of-plane direction at 1.77 A$^{-1}$ (the π–π stacking distance was 3.55 Å). The coherence length for these two crystalline planes were 7.49 and 2.92 nm estimated by Scherrer equation. Both MO-IDIC-2F and MO-IDIC showed face-on orientation and similar molecular packing. MO-IDIC-2F exhibits a (100) reflection at 0.39 A$^{-1}$ in the in-plane direction, with π–π stacking at 1.84 A$^{-1}$ in the out-of-plane direction. MO-IDIC shows a relatively weak (100) reflection at 0.41 A$^{-1}$ in the in-plane direction and π–π stacking at 1.82 A$^{-1}$ in the out-of-plane direction. The π–π stacking distance of these molecules are 3.41 Å for MO-IDIC-2F and 3.45 Å for MO-IDIC. The crystal coherence length for these two crystalline planes of MO-IDIC are 10.8 and 3.34 nm, respectively, while for MO-IDIC-2F, these two values are 14.35 and 3.75 nm. Besides, it should also be noted that the scattering intensity for MO-IDIC-2F in both (100) and π–π direction is stronger than that of MO-IDIC. These results imply that MO-IDIC-2F has a higher degree of self-organization and molecular packing than that of MO-IDIC. For the blend films (Fig. 4), the (100) peaks at 0.27 and 0.40 A$^{-1}$ in the in-plane direction came from donor and acceptor, respectively. The out-of-plane diffraction signal located at 1.80 A$^{-1}$ was a simple combination of (010) reflection of both donor and acceptor. In the in-plane line cut of blends, it was clear to see that the crystallinity of MO-IDIC-2F is

much better than MO-IDIC, for that the peak of MO-IDIC-2F at 0.40 A$^{-1}$ in the in-plane direction is sharper than that of MO-IDIC, which could also be observed in the GIWAXS result of neat film. Furthermore, crystal size of these two acceptors (MO-IDIC and MO-IDIC-2F) in blend were 6.4 and 10.8 nm, respectively, and this also accounted for that the efficiency of device based on MO-IDIC-2F is higher than that based on MO-IDIC. After thermal-annealing (Fig. 4f), the intensity of all three characteristic peaks increased, which meant the crystallinity of the system was improved and the arrangement of molecules became more ordered, which could also be proved by the improvement of crystal size. The improvement of molecular packing could be beneficial for the higher photovoltaic efficiencies of the PSCs with the thermal annealing treatment.

It is well known that the performance of PSCs is closely related to the film morphology. To investigate the difference of these two systems and the influence of thermal annealing, we measured their transmission electron microscopy (TEM) images (Supplementary Fig. 9). The as-cast blend of PTQ10:MO-IDIC shows a relatively homogeneous film, and after thermal annealing, small domain size about 50 nm appeared in the blend of PTQ10:MO-IDIC. For the thermal annealed blend film of PTQ10:MO-IDIC-2F, smaller domains and interpenetrating networks were formed, which benefits for exciton dissociation and the improvement of photovoltaic performance (especially $J_{sc}$ and FF).

Phase separation of the blends was further studied by resonant soft X-ray scattering (RSoXS). Figure 4g shows RSoXS scattering profiles using optimized photon energy (284.2 eV). There was no obvious peak, which could be due to that the acceptor had less contrast with polymer donor, thus structure feature could not be clearly observed. However, from the distribution of intensity, we could get some important information on the morphology. For this two-phase system, scattering curve could be treated using the Debye–Beuche equation $(I(q))^{-1/2} = K(a^3 Q)^{-1}(1 + a^2 q^2)$, where $K$ is a constant, $a$ is the correlation length, and $Q = \phi_1 \phi_2 (b_1 - b_2)^2$ is the scattering invariant where $\phi_i$ is the volume fraction of phase $i$ with an X-ray scattering length density of $b_i$. By fitting the data in the low $q$ region, we could get the correlation length of 21.4 and 26.1 nm for the thermal annealed blend films of PTQ10: MO-IDIC-2F, and PTQ10:MO-IDIC, respectively. If we assume that the volume fraction $\phi_i$ of the phases was equal to the volume fraction of the components, then the average sizes of each domain can be calculated to be about 42.8 and 52.2 nm, respectively, which was consistent with the TEM results.

**Cost analysis**. These two n-OS acceptors can be synthesized in high yields with relatively less synthetic steps in comparison with other acceptors. In order to further investigate and analyze the cost of the materials used in the active layers of PSCs, we built a model (see Methods) and present a detailed quantitative cost calculations based on the synthetic procedures and evaluation rules published in literatures[60–62] (Supplementary Table 8). In this model of the cost calculation, all the raw materials, solvent and reagent used for the reaction or purification were taken into account in order to estimate the total material costs. However, the energy input, either heating or cooling, was ignored to simplify the calculations. Therefore, the calculations are unlikely to reflect the actual cost of mass production, but can be used as a rough indication of synthetic complexity. In the calculation, 19 high performance donor and acceptor materials were considered with the production scale of 100 g. The materials include the n-OS acceptors of IDIC[42], ITIC[41], ITIM[63], ITIC-4F[23], NITI[35], MO-IDIC-2F, and C8-ITIC[34], as well as the polymer donors of PBDB-T[35], PBDB-T-SF[23], PFDBD-T[34], PBTA-TF[63], PBDTS-TDZ[64], PDTB-EF-T(P2)[65], PBDB-T-2Cl[33], PBDB-T-2F[66], and PTQ10[40],

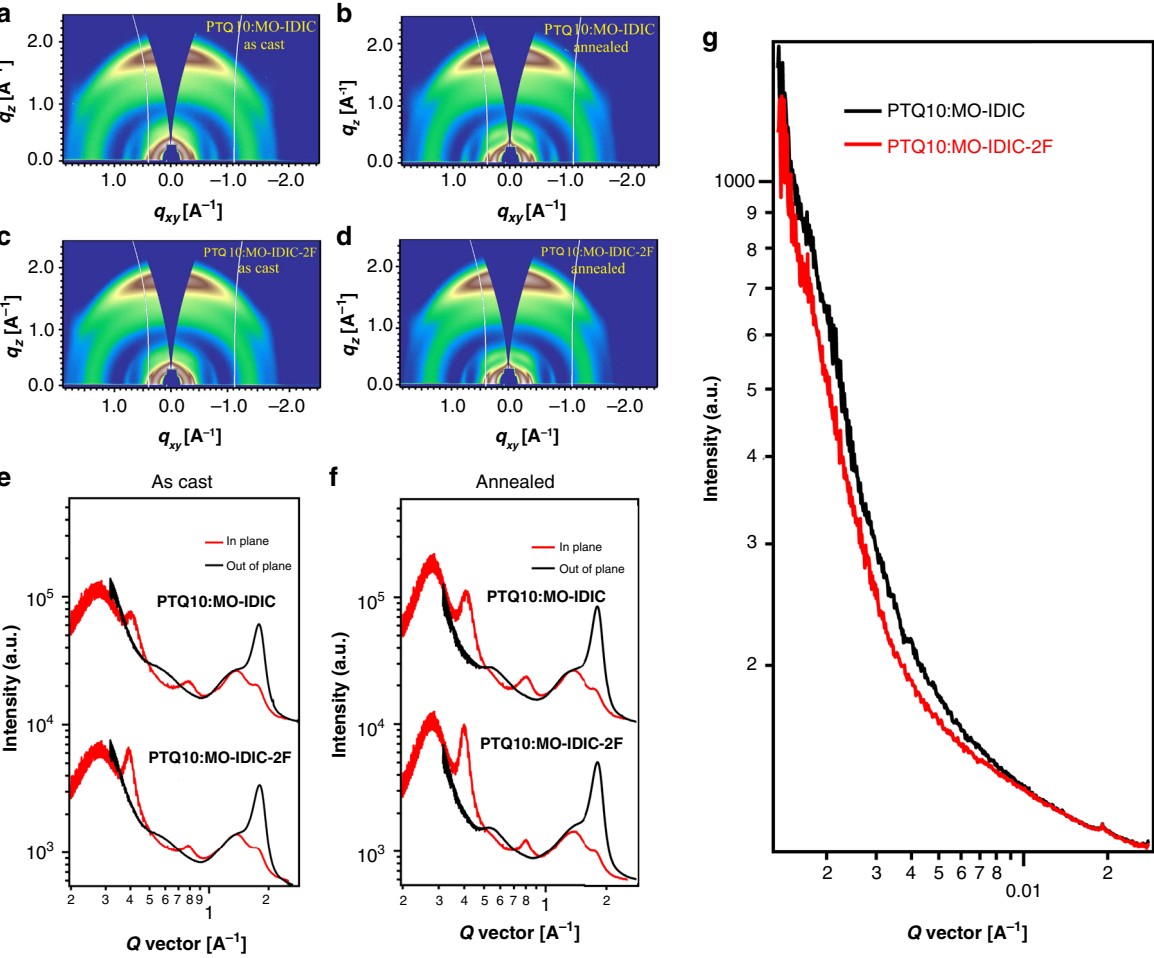

**Fig. 4** GIWAXS and RSoXS results of active layers of the PSCs. 2D GIWAXS patterns of **a** as cast PTQ10/MO-IDIC blend films; **b** PTQ10/MO-IDIC blend films with thermal annealing at 120 °C for 5 min; **c** as cast PTQ10/MO-IDIC-2F blend films; **d** PTQ10/MO-IDIC-2F blend films with thermal annealing at 110 °C for 5 min; **e** line cuts of GIWAXS images of the as cast PTQ10/acceptors blend films; **f** line cuts of GIWAXS images of the annealed PTQ10/ acceptors blend films; and **g** the RSoXS profiles of PTQ10/MO-IDIC and PTQ10/MO-IDIC-2F blend films as cast and thermal-annealed

each of these materials has demonstrated PCE higher than 12.5% in the PSCs. Meanwhile, MO-IDIC, O-IDTBR[67] and P3HT were chosen as comparison. The detailed statistic process for this model was attached in supplementary information in Supplementary Tables 8–10, and the results are summarized in Supplementary Table 11. It can be seen from Supplementary Table 11 that both MO-IDIC-2F acceptor and PTQ10 donor are relatively lower cost materials. In combination of their high photovoltaic performance, the PSCs based PTQ10:MO-IDIC-2F are promising for future commercial application.

Figure 5a displays the plots of material cost (¥ g$^{-1}$) versus synthesis steps for the high performance photovoltaic materials (extracted from their synthetic routes in Supplementary Figures 10–26), and the corresponding statistical data are listed in Supplementary Table 11. It can be seen that the cost of materials ($C_g$, cost-per-gram) is basically linear growth with the number of synthetic steps. Most of these materials need more than seven steps with cost higher than 250¥ g$^{-1}$. In this work, by optimizing synthetic procedures, MO-IDIC and MO-IDIC-2F exhibit the lowest cost than other acceptors and MO-IDIC with the minimum synthesis steps of six steps present only 173.8¥ g$^{-1}$, which is close to the cost of P3HT. Above all, the results indicate the importance of optimizing the synthetic procedures to improve the yield and minimize the number of individual synthetic steps.

Of course, using cheap raw materials would be also beneficial to achieve low cost photovoltaic materials.

In order to further investigate the effect of photovoltaic material costs on the commercial availability of PSCs, $C_w$ (cost-per-peak-Watt, ¥ W$_p$$^{-1}$) was introduced as an evaluation parameter[38,57] for the cost of the photovoltaic materials of the PSCs. $C_w$ is calculated using the following equation: $C_w = (C_{g\ total} \times \rho \times t)/(\eta \times I)$, where $C_{g\ total}$ is the total material costs of active layer for a PSC device ($C_{g\ total} = 0.5(C_{g\ donor} + C_{g\ acceptor})$ in considering that most of the optimized D/A weight ratios in the active layer of the PSCs are 1:1); $\rho$ is the density of the materials (here using 1.1 g cm$^{-3}$); $t$ is the thickness of the active layer with the unit of 100 nm; $\eta$ is the PCE value of the PSCs, and $I$ is the solar insolation under peak conditions (assumed to be 1000 W m$^{-2}$). The calculated data are listed in Table 2. For the PSCs based on PTQ10:MO-IDIC with a PCE of 11.2%, its active layer possesses the lowest cost of 216.5¥ g$^{-1}$, and its $C_w$ value of 0.193¥ W$_p$$^{-1}$ is also relatively low. For the PTQ10:MO-IDIC-2F-based devices (as seen in Fig. 5b), the $C_{g\ total}$ of its active layer is 254.9¥ g$^{-1}$ which is a little higher than that of MO-IDIC (216.5¥ g$^{-1}$), however its higher PCE of 13.46% leads to a decreased $C_w$ value of 0.190¥ W$_p$$^{-1}$. For the P3HT-based PSCs with O-IDTBR as acceptor, the relatively low PCE (7%)[67] results in higher $C_w$ value (0.538¥ W$_p$$^{-1}$). In contrast, other high performance PSCs show much higher $C_w$ values

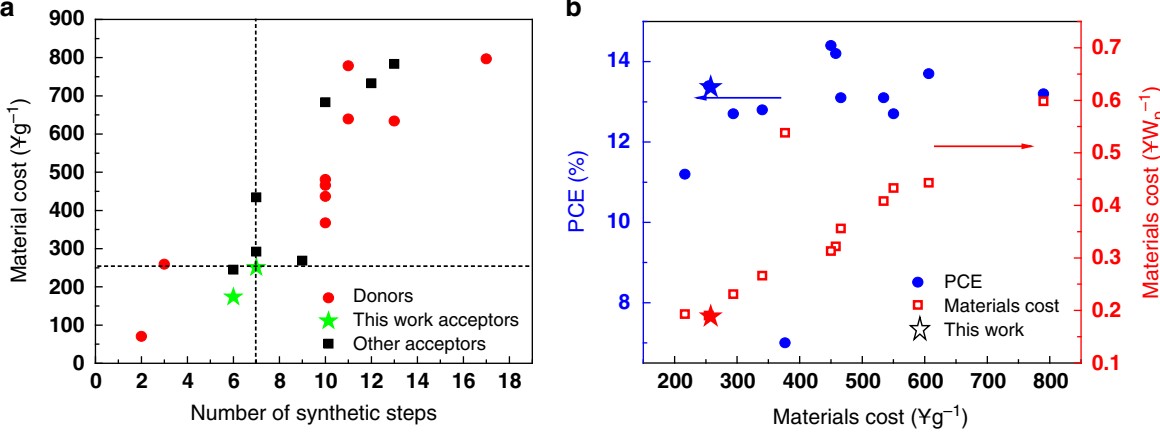

**Fig. 5** Cost analysis of the photovoltaic materials. **a** Plot of the calculated material costs ($¥\,g^{-1}$) versus the number of required synthetic steps for the high performance photovoltaic materials and **b** plots of PCE or calculated material cost-per-peak-Watt ($¥\,W_p^{-1}$) versus the materials cost-per-gram ($¥\,g^{-1}$)

### Table 2 Survey of the donor and acceptor materials, as well as the maximum PCE and material costs $C_g$ and $C_w$

| Materials | | PCE (%) | $C_g$ ($¥\,g^{-1}$) | $C_w$ ($¥\,W_p^{-1}$) | Reference |
|---|---|---|---|---|---|
| **Donor** | **Acceptor** | | | | |
| PTQ10 | MO-IDIC-2F | 13.4 | 254.9 | 0.190 | This work |
| PTQ10 | MO-IDIC | 11.2 | 216.5 | 0.193 | This work |
| PTQ10 | IDIC | 12.7 | 293.9 | 0.231 | 40 |
| PBDTS-TDZ | ITIC | 12.8 | 340.4 | 0.266 | 64 |
| PBTA-TF | ITIM | 13.1 | 465.9 | 0.356 | 63 |
| PBDB-T-SF | ITIC-4F | 13.1 | 534.3 | 0.408 | 23 |
| PBDB-T | NITI | 12.7 | 550.1 | 0.433 | 35 |
| PFDBD-T | C8-ITIC | 13.2 | 789.9 | 0.598 | 34 |
| PDTB-EF-T(P2) | ITIC-4F | 14.2 | 457.8 | 0.322 | 65 |
| PBDB-T-2Cl | ITIC-4F | 14.4 | 450.2 | 0.313 | 33 |
| PBDB-T-2F | ITIC-4F | 13.7 | 606.6 | 0.443 | 66 |
| P3HT | O-IDTBR | 7.0 | 376.7 | 0.538 | 67 |

(Table 2) due to the higher cost of the photovoltaic materials. Overall, the devices based on PTQ10: MO-IDIC-2F possess the higher PCE and the lowest $C_w$ among the high performance PSCs, demonstrating great potential for the commercial application of PSCs.

## Discussion

A new synthetic route was developed for simplifying the synthetic processes of high performance *n*-OS acceptors with alkyl side chains. By this synthetic route, the synthetic step of IDIC was reduced. Moreover, by introducing alkoxy substituents on the benzene unit of the fused ring core, two new *n*-OS acceptors of MO-IDIC and MO-IDIC-2F were synthesized with further simplified synthetic processes and higher overall yield. The two acceptors possess broad absorption and higher electron mobilities, MO-IDIC-2F shows red-shifted absorption and higher degree of self-organization than that of MO-IDIC. The PCE values of the as-cast PSCs based on PTQ10:MO-IDIC and PTQ10:MO-IDIC-2F reached 10.76% and 12.39%, respectively. The PCE of 12.39% is the highest efficiency for the PSCs without post treatment reported in literatures so far. After thermal annealing at a relatively low temperature of 110 °C for 5 min, PCE of the PTQ10:MO-IDIC-2F-based PSCs was further improved to 13.46%. Meanwhile, its photovoltaic performance is insensitive to the active layer thickness between 100 and 300 nm. The PCE of the PSCs with the active layer thickness of 250 and 300 nm still reached 12.63% and 11.01%, respectively, which is conducive to

the large area fabrication of PSCs. Based on the cost analysis of the PSCs, the PSCs based on PTQ10:MO-IDIC-2F shows great advantages of low cost, high photovoltaic performance and thickness-insensitivity in comparison with the acceptors with PCE over 12.5%. The results indicate that MO-IDIC-2F is a promising low cost acceptor for commercial application of PSCs.

## Methods

**Materials and synthesis**. All chemicals and solvents were purchased from Innochem, J&K, Alfa Aesar, and TCI Chemical Co., respectively. The solvents do not need to degas in all the reactions, and the reactions all performed under a nitrogen atmosphere. Compounds **5**, **6** and PTQ10 were synthesized according to the procedures reported in the literatures[29,49,65]. The synthetic route of MO-IDIC and MO-IDIC-2F are shown in Figs. 1, 2 and their detailed synthesis processes are described in the following.

**Synthesis of Compounds 3 and 3'**. Benzene-1,4-diboronic acid bispinacol ester (1.65 g, 5 mmol) or Compound **5** (1.95 g, 5 mmol), Compound **6** (2.56 g, 11 mmol), Pd(OAc)$_2$ (56 mg, 0.25 mmol), and tBu$_3$PHBF$_4$ (144 mg, 0.5 mmol) were dissolved in acetone (60 mL) under a nitrogen atmosphere. After stirred for 5 min, 8 mL NaOH aqueous solution (2 M) was added, and the reaction mixture was stirred for 1 h, then the mixture was extracted with dichloromethane (100 mL × 3) and water (100 mL). The collected organic layer was dried over MgSO$_4$. After removal of the solvent under reduced pressure, the residue was purified by washing with methanol and petroleum ether to give a white product **3** or **3'**.

**3** (1.75 g, 91% yield). $^1$H NMR (400 MHz, CDCl$_3$) $\delta$ 7.55–7.50 (m, 6H), 7.26 (d, $J$ = 5.3 Hz, 2H), 4.23 (q, $J$ = 7.1 Hz, 4H), 1.22 (t, $J$ = 7.1 Hz, 6H). $^{13}$C NMR (101 MHz, CDCl$_3$) $\delta$ 163.28, 150.11, 133.61, 130.11, 129.40, 128.58, 124.25, 60.57, 14.04. MS (EI$^+$) $m/z$ calcd. for [M]$^+$ C$_{20}$H$_{18}$O$_4$S$_2$ 386, found 386.

**3'** (2.05 g, 92% yield). $^1$H NMR (300 MHz, CDCl$_3$) $\delta$ 7.50 (d, $J$ = 5.3 Hz, 2H), 7.29 (d, $J$ = 5.3 Hz, 2H), 6.92 (s, 2H), 4.18 (q, $J$ = 7.1 Hz, 4H), 3.73 (s, 6H), 1.17 (t, $J$

= 7.1 Hz, 6H). [13]C NMR (75 MHz, CDCl$_3$) δ 163.57, 150.43, 144.81, 130.81, 129.25, 124.45, 123.72, 114.42, 60.38, 56.23, 14.02. MS (EI$^+$) $m/z$ calcd. for [M]$^+$ C$_{22}$H$_{22}$O$_6$S$_2$ 446, found 446.

**Synthesis of compound 4 and 4′.** Compound **3** (1.93 g 5 mmol) or **3′** (2.23 g 5 mmol) was dissolved in dry THF (60 mL) and placed under a nitrogen atmosphere. The solution was cooled to 0 °C and stirred while 32 mL hexylmagnesium bromide (0.8 M) was added dropwise. The mixture was warmed to room temperature and stirred for 12 h. It was then poured into water and extracted with dichloromethane. The organic extracts were dried over anhydrous MgSO$_4$. After removal of the solvent, the residue was purified by silica gel chromatography (1:1, Hexanes:DCM) to give white solid **4** or **4′**.

**4** (2.71 g, 85%) [1]H NMR (400 MHz, CDCl$_3$) δ 7.39 (s, 4 H), 7.23 (d, $J$ = 5.3 Hz, 2H), 6.98 (d, $J$ = 5.2 Hz, 2H), 1.75–1.58 (m, 10H), 1.38–1.09 (m, 32H), 0.85 (t, $J$ = 6.4 Hz, 12H). [13]C NMR (75 MHz, CDCl$_3$) δ 143.15, 136.43, 135.27, 130.00, 128.08, 128.03, 123.65, 43.37, 31.84, 29.63, 23.67, 22.66, 14.10. HRMS (TOF) $m/z$ calcd. for [M + Na]$^+$ C$_{40}$H$_{62}$NaO$_2$S$_2$ 661.4089, found 661.4078.

**4′** (3.07 g, 88%) [1]H NMR (400 MHz, CDCl$_3$) δ 7.28 (d, $J$ = 5.3 Hz, 2H), 6.94 (d, $J$ = 5.3 Hz, 2H), 6.83 (s, 2H), 3.72 (s, 6H), 2.86 (s, 2H), 1.85-1.57 (m, 8H), 1.34–1.11 (m, 32H), 0.85 (t, $J$ = 6.7 Hz, 12H). [13]C NMR (75 MHz, CDCl$_3$) δ 149.66, 143.60, 131.79, 126.87, 124.57, 123.43, 114.93, 55.29, 41.99, 31.15, 28.97, 22.92, 21.85, 13.26. HRMS (TOF) $m/z$ calcd for [M]$^+$ C$_{42}$H$_{66}$O$_4$S$_2$ 698.4403, found 698.4396.

**Synthesis of Compound 2 and MO-IDT.** Compound **4** (1.98 g, 3 mmol) or **4′** (2.09 g, 3 mmol) was dissolved in dry toluene (30 mL) and placed under a nitrogen atmosphere, then amberlyst15 (2 g) as catalyst (Acros Amberlyst15, (dry) ion-exchange resin) was added and heated at 85 °C for 12 h. After the reaction, the mixture was filtered and the organic liquids were collected. The catalyst was washed with dichloromethane for recycle (the catalyst was cleaned in distilled water, dilute hydrochloric acid, distilled water, and ethanol in sequence, then drying). After removal of the solvent under reduced pressure, the residue was purified by column chromatography on silica gel using petroleum ether as eluent to give white solid **2** (from Compound **4**) or **MO-IDT** (from Compound **4′**).

**2** (1.14 g, 63% yield).[1]H NMR (400 MHz, CDCl$_3$) δ 7.27 (s, 2H), 7.25 (d, $J$ = 4.8 Hz, 2H), 6.96 (d, $J$ = 4.8 Hz, 2H), 2.03–1.79 (m, 8H), 1.20–1.02 (m, 24H), 0.85–0.73 (m, 20H). [13]C NMR (101 MHz, CDCl$_3$) δ 155.10, 153.23, 141.66, 135.59, 126.13, 121.72, 113.13, 53.67, 39.21, 31.62, 29.73, 24.15, 22.60, 14.05. HRMS (TOF) $m/z$ calcd for [M]$^+$ C$_{40}$H$_{58}$S$_2$ 602.3980, found 602.3983.

**MO-IDT** (1.83 g, 92% yield). [1]H NMR (300 MHz, CDCl$_3$) δ 7.31 (d, $J$ = 4.8 Hz, 2H), 6.93 (d, $J$ = 4.8 Hz, 2H), 3.97 (s, 6H), 2.26-1.97 (m, 8H), 1.12-1.06 (m, 24H), 0.80–0.72 (m, 20H). [13]C NMR (75 MHz, CDCl$_3$) δ 152.64, 143.69, 143.16, 135.55, 128.86, 125.29, 118.92, 59.14, 53.36, 36.02, 29.64, 27.58, 22.22, 20.52, 11.95. HRMS (TOF) $m/z$ calcd. for [M]$^+$C$_{42}$H$_{62}$O$_2$S$_2$ 662.4182, found 662.4185.

**Synthesis of Compound 7.** A Vilsmeier reagent, which was prepared with POCl$_3$ (0.62 mL, 6.4 mmol) in DMF (2.00 mL, 25.84 mmol), was added to a cold solution of Compound MO-IDT (212 mg, 0.32 mmol) in dry CHCl$_3$ (20 mL) at 0 °C under a nitrogen atmosphere. After being stirred at 60 °C for 12 h, the mixture was poured into ice water (100 mL), neutralized with Na$_2$CO$_3$, and then extracted with dichloromethane. The combined organic layer was washed with water and brine, dried over anhydrous MgSO$_4$. After removal of solvent, it was purified by column chromatography on silica gel using petroleum ether/dichloromethane (1:1) as eluent, yielding a yellow solid **7** (196 mg, 85% yield). [1]H NMR (400 MHz, CDCl$_3$) δ 9.93 (s, 2H), 7.61 (s, 2H), 4.00 (s, 6H), 2.28–2.04 (m, 8H), 1.15–1.08 (m, 24H), 0.80–0.73 (m, 20H). [13]C NMR (75 MHz, CDCl$_3$) δ 182.48, 155.05, 146.53, 146.42, 145.93, 131.67, 128.85, 61.00, 55.36, 37.21, 30.95, 28.86, 23.73, 21.90, 13.34. HRMS (TOF) $m/z$ calcd. for [M]$^+$ C$_{44}$H$_{62}$O$_4$S$_2$ 718.4079, found 718.4084.

**Synthesis of MO-IDIC.** Compound **7** (145 mg, 0.2 mmol) and 1,1-dicyano-methylene-3-indanone (194 mg, 1 mmol) was dissolved in CHCl$_3$ (25 mL) under a nitrogen atmosphere. 0.6 mL pyridine was added and refluxed for 12 h. Then, the mixture was poured into water (100 mL) and extracted with CHCl$_3$ (2 × 100 mL). The organic layer was washed with water, and then dried over MgSO$_4$. After removing the solvent, the residue was purified using column chromatography on silica gel employing petroleum ether/CHCl$_3$ (1:4) as an eluent, yielding a dark blue solid MO-IDIC (189 mg, 88%). [1]H NMR (400 MHz, CDCl$_3$) δ 8.97 (s, 2H), 8.72 (d, $J$ = 6.8 Hz, 2H), 7.96 (d, $J$ = 8.1 Hz, 2H), 7.83–7.69 (m, 6H), 4.13 (s, 6 H), 2.37–2.08 (m, 8H), 1.23–0.95 (m, 24H), 0.81–0.72 (m, 20H). [13]C NMR (75 MHz, CDCl$_3$) δ 188.11, 160.65, 157.17, 156.11, 147.97, 147.79, 142.07, 139.92, 138.27, 136.90, 136.77, 134.99, 134.39, 134.11, 125.25, 123.66, 122.16, 114.75, 114.69, 68.84, 61.84, 55.94, 37.97, 31.49, 29.39, 24.36, 22.45, 13.89. HRMS (TOF) $m/z$ calcd. for [M]$^+$ C$_{68}$H$_{70}$N$_4$O$_4$S$_2$ 1070.4833, found 1070.4833.

**Synthesis of MO-IDIC-2F.** Compound **7** (145 mg, 0.2 mmol) and 2-(5 or 6-difluoro-3-oxo-2,3-dihydro-1H-inden-1-ylidene) malononitrile (300 mg, 1.4 mmol) were dissolved in CHCl$_3$ (25 mL) under a nitrogen atmosphere. 0.6 mL pyridine was added and refluxed for 12 h. Then, the mixture was poured into water (100 mL) and extracted with CHCl$_3$ (2 × 100 mL). The organic layer was washed with water, and then dried over MgSO$_4$. After removing the solvent, the residue

was purified using column chromatography on silica gel employing petroleum ether/CHCl$_3$ (1:4) as an eluent, yielding a dark blue solid MO-IDIC-2F (203 mg, 92%).[1]H NMR (400 MHz, CDCl$_3$) δ 8.96 (d, $J$ = 3.4 Hz, 2H), 8.74 (dd, $J$ = 8.7, 4.2 Hz, 1H), 8.40 (dd, $J$ = 9.0, 2.0 Hz, 1H), 7.96 (dd, $J$ = 8.3, 5.2 Hz, 1H), 7.75 (d, $J$ = 4.2 Hz, 2H), 7.59 (dd, $J$ = 6.7, 2.5 Hz, 1H), 7.47–7.38 (m, 2H), 4.13 (s, 6H), 2.36–2.08 (m, 8H), 1.16–1.08 (m, 24H), 0.80–0.73 (m, 20H).[13]C NMR (101 MHz, CDCl$_3$) δ 186.70, 167.99, 167.66, 165.42, 165.06, 159.59, 159.31, 157.41, 156.70, 156.55, 148.15, 147.94, 142.34, 142.24, 142.11, 138.54, 138.41, 137.07, 135.85, 134.28, 133.20, 127.89, 127.78, 125.92, 125.82, 122.12, 122.05, 121.92, 121.69, 114.75, 114.65, 114.49, 114.29, 112.94, 112.68, 110.89, 110.66, 69.74, 68.76, 61.92, 56.05, 38.03, 31.55, 29.45, 24.44, 22.51, 13.94. HRMS (TOF) $m/z$ calcd. for [M]$^+$ C$_{68}$H$_{68}$F$_2$N$_4$O$_4$S$_2$ 1106.4645, found 1106.4644.

**Measurements.** [1]H NMR spectra were measured on a Bruker DMX-400 spectrometer with $d$-chloroform as the solvent and trimethylsilane as the internal reference. UV–visible absorption spectra were measured on a Hitachi U-3010 UV–vis spectrophotometer. Mass spectra measurement was performed on a Shimadzu spectrometer. TGA was conducted under a nitrogen flow rate of 100 mL min$^{-1}$ on a Perkin-Elmer TGA-7 thermogravimetric analyzer at a heating rate of 20 °C min$^{-1}$. The electrochemical cyclic voltammetry was carried out on a Zahner IM6e Electrochemical Workstation, in an acetonitrile solution of 0.1 mol L$^{-1}$ $n$-Bu$_4$NPF$_6$ at a potential scan rate of 100 mV s$^{-1}$. The sample film on Pt plate was used as working electrode, a platinum wire was used as counter electrode and Ag/AgCl was used as reference electrode. An atomic force microscope (AFM, SPA-400) with the tapping mode was used to measure the film morphology.

**Device fabrication and characterization.** The PSCs were fabricated with a structure of ITO/PEDOT:PSS (40 nm)/active layer/PDINO/Al. A thin layer of PEDOT:PSS was deposited on precleaned ITO-coated glass through spin-coating a PEDOT:PSS aqueous solution (Baytron P VP AI 4083 from H.C. Starck) at 4000 rpm and dried subsequently at 150 °C for 15 min in air. Then the PEDOT:PSS-coated ITO glass was transferred to a nitrogen glove box, where the active blend layer of PTQ10 donor and acceptors was prepared by spin-coating their chloroform solution onto the PEDOT:PSS layer at a spin-coating rate of 3000 rpm. Then the active layers were annealed at 110 or 120 °C for 5 min for the devices with thermal annealing treatment. PDINO was synthesized in our lab according to the procedures reported in the literature[68]. The methanol solution of PDINO with a concentration of 1.0 mg mL$^{-1}$ was spin-coated atop the active layer at 3000 rpm for 30 s to form a PDINO cathode buffer layer with thickness of ca. 10 nm. Finally, top Al electrode was deposited in vacuum onto the cathode buffer layer at a pressure of ca. $5.0 \times 10^{-5}$ Pa. The active area of the PSCs was 4.7 mm$^2$, which was defined by Optical microscope (Olympus BX51). The current density–voltage ($J$–$V$) curves of the PSCs were measured on Keithley 2450 Source-Measure Unit in a glove box filled with nitrogen (oxygen and water contents are smaller than 0.1 ppm). And the measurements were performed by scanning voltage from −1.5 to 1.5 V with a voltage step of 10 mV and delay time of 1 ms. Oriel Sol3A Class AAA Solar Simulator (model, Newport 94023 A) with a 450 W xenon lamp and an air mass (AM) 1.5 filter was used as the light source, and the light intensity was calibrated to 100 mW cm$^{-2}$ by a Newport Oriel 91150V reference cell. In order to accurately measure the photocurrent, mask with an area of 2.2 mm$^2$ was used to define the effective area of the PSCs. The devices with or without mask showed consistent photovoltaic performance values with relative errors within 0.5%. The photovoltaic parameters in this work are measured from the devices without mask and the PCE statistics were calculated using more than 20 individual devices fabricated under the same conditions. Solar Cell Spectral Response Measurement System QE-R3-011 (Enli Technology Co., Ltd., Taiwan) was used to measure the IPCE. The light intensity at different wavelength was calibrated with a standard single-crystal Si photovoltaic cell.

**Mobility measurements.** Charge carrier (hole and electron) mobilities were measured with the SCLC method. Hole-only devices with the structures of ITO/PEDOT:PSS/PTQ10: acceptor (1:1, w/w)/Au () was used to measure the hole mobility and electron-only devices with the structure of ITO/ZnO/PTQ10:acceptor (1:1, w/w)/PDINO/Al was used to measure the electron mobility. The mobility values were calculated by MOTT-Gurney equation:[22]

$$J = \frac{9\varepsilon_r\varepsilon_0\mu V^2}{8L^3} \qquad (1)$$

Herein we use a relative dielectric constant of 4 for $\varepsilon_r$, and the built-in voltage $V_{bi}$ are 0.2 and 0 V, respectively, for the hole-only and the electron-only devices.

**GIWAXS characterization.** GIWAXS measurements were conducted at Advanced Light Source (ALS), Lawrence Berkeley National Laboratory, Berkeley, CA at the beamline 7.3.3. Data was acquired at the critical angle (0.16°) of the film with a hard X-ray energy of 10 keV. X-ray irradiation time was 30–60 s, dependent on the saturation level of the detector. The scattered intensity was detected with a Pilatus detector. 1D profile was obtained with the intensity distribution analyzed along in-plane and out-of-plane direction. It should be noted that the region at low $q$

position was blocked by beamstop, and thus no signal could be observed in 1D profile at the low q position. Crystal coherence lengths (CCL) are estimated based on the Scherrer equation ($L = 2\pi K/\text{FWHM}$), where $K$ is the shape factor (here we use 0.9), and FWHM is the full width at half maximum of diffraction peaks.

**Quantitative analysis of the material costs**. In order to calculate the material costs in active layer of the PSCs, we built a simplified model to quantitatively analyze the synthetic cost of the materials. In this model, the cost of raw materials, solvent, and consumption in purification were included, the energy input was eliminated for simplifying the model. The model is based on published small-scale synthetic procedures (Supplementary Tables 8–10), however, the amount of prepared compounds we choose is 100 g, so an important assumption in this system is that the prices for the starting materials, reagents, and solvents will remain invariant. Actually, the industrial production of materials may result in a considerable reduction in raw material costs, thus, the costs of the photovoltaic materials which used in active layer may be further reduced under the condition of large-scale industrial production. In this model, all the price data of chemicals and solvents come from Innochem, J&K, Solarmer Materials Inc, Derthon Optoelectronic Materials Science Technology Co Ltd., Alfa Aesar, TCI Chemical Co. etc.

In this model, the research samples include the acceptors of IDIC, O-IDTBR, ITIC, ITIM, ITIC-4F, NITI, MO-IDIC-2F, MO-IDIC, C8-ITIC and the polymers donors of PBDB-T, PBDB-T-SF, PFDBD-T, PBTA-TF, PBDTS-TDZ, PDTB-EF-T (P2), PBDB-T-2Cl, PBDB-T-2F, PTQ10, P3HT. Based on the published synthetic procedures of these molecules, we chose synthetic routes with the highest overall yields if there are multiple reports on the synthetic methods and the starting materials are simple molecules that are currently available from commercial companies. The detailed synthesis process for all the materials are shown in Supplementary Figs. 10–26. In these figures, synthetic steps, raw materials, reagents, yield per step were included and isolation or purification steps were numerically labeled. The isolation and purification of products from the crude mixture have the important effect on costs of compounds, which involve considerably the material inputs and waste outputs. In order to ensure a fair comparison for different synthetic process, we provide a standard for induction and specification of the isolation or purification steps, which include 1. quenching/neutralization; 2. filtration; 3 extraction; 4. column chromatography; 5. recrystallization; and 6. distillation. The details for each procedure are based on standard organic laboratory techniques and are summarized below:

Quenching/neutralization: This process was evaluated on a case-by-case basis. One-to-one molar equivalents of acid, base, or water were assumed to neutralize reactive intermediates or side-products, this is a very small part of purification.

Filtration: We assume 1 g product needs 100 mL water. However, in considering the low cost of water in comparison with the organic solvent and reaction reagents, the amount of water is not counted in the cost calculation for simplifying the model.

Extraction: 150 mL (three individual 50 mL extractions) solvent (dichloromethane used as the most commonly extraction solvent) and 1 g of a drying agent ($MgSO_4$) is assumed to be necessary to extract 1 g of crude product.

For the polymers, the mixture need Soxhlet extraction, and we assume 150 mL methanol, 150 mL hexanes, and 150 mL chloroform were used for the purification of 1 g polymer, then the chloroform fraction was reduced and dried in vacuo.

Column chromatography: An ideal separation is assumed, and petroleum ether/dichloromethane are used as eluent. We assume 4 L of effluent and 600 g $SiO_2$ particles (a column that is 45 cm long and 50 mm in diameter) are used for the purification of 5 g sample.

Recrystallization: We assume that 1 g product needs 100 mL solvent and the procedure is performed only once.

Distillation: We assume no solvent or chemical waste, only energy input. We neglected the energy input in the cost calculation for simplifying the model. Actually, the distillation was used only in the first step of preparing PBDB-T and PBDB-T-2Cl, which have little impact on this statistical calculation.

The detailed synthetic routes of the materials are shown in Supplementary Figs. 10–26 and statistical data are summarized in Supplementary Tables 8–10.

## Data availability

The data that support the findings of this study are available from the corresponding author upon reasonable request.

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

## Acknowledgements

This work was supported by NSFC (Nos. 91633301, 21734008, 51820105003, and 51673200) and the Strategic Priority Research Program of the Chinese Academy of Sciences, Grant No. XDB12030200. T.P.R. thanks the support by the U.S. Office of Naval Research under contract N00014-15-1-2244. Portions of this research were carried out at beamline 7.3.3 and 11.0.1.2 at the Advanced Light Source, Molecular Foundry, and National Center for Electron Microscopy, Lawrence Berkeley National Laboratory, which was supported by the DOE, Office of Science, and Office of Basic Energy Sciences.

## Author contributions

X.L. designed the synthetic routes, synthesized, and characterized the *n*-OS acceptors. F. P. carried out the PSCs fabrication and characterization. C.S. provided PTQ10 polymer donor, J.D. participated in the discussion of the synthesis. L.X. and Z.Z. provided the cathode buffer layer material. M.Z., J.W., and F.L. measured the GIWAXS diffraction patterns, Z.W., M.X., and C.Z. measured the transient spectra, Y.L. supervised the project. X.L. and Y.L. wrote the paper.

## Additional information

**Competing interests:** The authors declare no competing interests.

