## [Peer Review File · Nature Communications]

Reviewers' comments:

Reviewer #1 (Remarks to the Author):

In this manuscript, the authors presented a new synthetic route for IDIC core with high yield. The authors further modified the core structure and achieved a high PCE of 13.5% with proper optimization. The authors also compared the cost of different donor and acceptors materials, they suggested that the cost of active material combination reported in this manuscript is the lowest among the high performance OPVS. This paper is clearly written and it could be recommended for publication after minor revisions:

1. Although the authors presented a new synthesis route to the IDIC core, the synthesis procedure for the key intermediate 4 and 4' are not clear enough. For example, the amount of the reactant, solvent and catalyst were not given. Besides, the authors should also provide the NMR and high-resolution mass spectra of 4 and 4' to verify the structure.

2. An interesting point is that most of the ring-closing conditions except amberlyst 15 failed to yield the desirable product (especially, the chemical structure of TsOH is very similar to amberlyst 15). Do the authors have any explanation on this? Besides, the source and pre-treatment of the catalyst was not included in the supporting information.

3. The alkoxy groups on the core are usually regarded as strong electron-donating units, which should induce stronger ICT and thus result in red shift of the absorption (Energy Environ. Sci., 2017, 10, 1610-1620). The authors should explain why in this case the absorption of MO-IDIC are almost the same as IDIC.

4. Detailed quantitative cost calculations on a variety of OPV materials was done by the authors to evaluate the industrial potential of different materials. However, we believe that stability issue is another problem that limited the commercialization of organic solar cells. In order to further improve the quality of this manuscript, the authors should provide the stability data of the devices.

5. The authors should include several important works in the development of highly efficient organic solar cells like: Nature Energy, (2016), 1, 15027, Nature Energy, (2016), 1, 16089, and "Nature Energy,(2018),3,720-731, DOI: 10.1038/s41560-018-0181-5

Reviewer #2 (Remarks to the Author):

Summary:

In this manuscript, Li et al. synthesized a new electron donating fused ring core, called MO-IDT, which is the previously published C6-IDIC core with a new methoxy substituent on the core benzene unit. They paired this new core with both INCN and mono fluorinated INCN acceptor pendants to make two fused-ring electron acceptors of MO-IDIC and MO-IDIC-2F. These acceptors yielded PCEs between 10-13% when paired with a previously reported PTQ-10 polymer donor. The authors highlight two major features of this material. First, the synthesis of the donor core has been redesigned to offer a higher yield, and second, because of this design, the PTQ-10:Acceptor blends offer a lower cost compared to other blends.

As a whole, this manuscript highlights good improvements in the synthesis of low cost materials, but the some of the details are superfluous and distract from the key themes of the work. The design of materials is interesting and should be published, but there is a variety of issues with the manuscript

that should be addressed first. It is indeed important to understand the cost and scalability of materials for OPVs, as this will become major hurdles in the transition from lab scale devices to commercial applications, so papers that include details are very important for the OPV field. While this manuscript highlights new materials that are lower cost, I believe it is best for a more specialized OPV focused journal rather than Nature Comm. In fact, this work might have more impact in a different journal that OPV researchers AND manufacturers would read.

Major Comments:

- **Partial Cost Analysis:** Throughout the text, the authors claim that their new synthetic process has lower energy consumption, but this is only conjecture as they provide no evidence to support their claim. As the authors included a partial cost analysis section, it is straightforward to complete this analysis fully and include energy use throughout (I also think there are other components such as hazardous waste production differences that would strengthen their claims). This is even more important as the authors make claims about energy consumption throughout the manuscript. There are a few earlier papers on this cost-analysis topic, which can help the authors strengthen this part.
- **Inclusion of Extra Information:** The authors included a rather large section on femtosecond-resolved transient absorption studies. I am not questioning the value which TA measurements can offer, but for this manuscript, TA doesn't really fit in. Specifically, what do we learn from these TA results? The information learned from TA doesn't support the claims that the authors are making. Because there is no comparison to materials made without their synthesis, or information learned about low cost, this information is superfluous and distracts for the themes established. The novelty of this work should focus on the synthesis and low cost attributes; therefore, I would remove this section all together.
- **Comparing both Yield and Performance:** The authors claim that the inclusion of the methoxy helps reaction yields, but I think it is also important to compare the effects on device performance. It would be very valuable to compare IDIC and IDIC-2F throughout the text, especially in Figure 3. For example, if the methoxy group increases the yield but the device performance decreases, does the increase in yield provide more value than the decrease in performance, OR, does the device performance and the yield both increase. I think that this comparison is much more important that as cast vs annealed.
- **Materials included in Cost Analysis:** In the cost analysis section, the authors include P3HT, but in Table 2 of the main text, there is no blends containing P3HT. Additionally, there have been multiple works which have yielded PCEs greater than 14%. I think that it would be important to include both of these into your cost analysis to provide a more complete understanding. As P3HT is the standard of commercial products at the moment, therefore it is very important make this comparison.
- **Erroneous Reaction Mechanism:** The reaction mechanisms shown in Figure S1 are fundamentally incorrect, for example, the arrows show things such as carbocations attacking.

Missing Data/Evidence:

- **Statistics:** Why are there only statistics (average and standard deviations) for half of the blends in Table 1? Please include statistics for all blends (including SI tables). Please also include the notation for Table 1, maximum value, average value, and standard deviation.
- **Optimization of Reaction Conditions:** The authors claim that they explored different equivalences of BF₃OEt₂, reaction time, as well as a variety of different acid catalysts like H₂SO₄, TsOH, BBr₃, AlCl₃, etc. For each of these, they didn't see any improvements in the yield. Please provide evidence of this. In the SI, there should be tables highlighting the different conditions and resulting yields.
- **Optimization of Device Performance:** The authors claim that the devices where optimized for ratio, solvents, etc. Please provide evidence highlighting the various conditions as tables in the SI.

Minor Comments:

- Figure 2 has the incorrect name for MO-IDIC-2F.
- Additionally in Figure 2, the method for energy level determination (CV/electrochem) should be mentioned in the figure caption.
- The authors claim that their materials exhibit strong optical absorption from 550-700 nm. I think that this number should be refined to 600-700 nm, as there is very little absorbance at 550 nm.
- The figure caption for Figure 3 doesn't include the blend or conditions for Fig 3c and 3d, i.e. which electron acceptor and is it as-cast or annealed?
- There are a few sections which have incorrect grammar which needs to be addresses. I will include a few here, but this isn't intended to be a complete list. Page 1, line 4 of abstract. First sentence of introduction. First sentence on Page 5 is an incomplete sentence. There are multiple locations where the references come before the punctuation. etc.
- The quality of the equations and volume fractions for RSoXS discussion (page 17) are very low. Please replace with better text.
- The authors claim that PQT10:MO-IDIC-2F has a fiber like structure, but looking at Figure S7b, I disagree. I do not see a fiber-like structure from the TEM.
- Discussion on Page 20 should be Conclusion
- Page 26: the authors claim there model is based on published small-sacle synthetic procedureds, but they do not provide references/citations for these procedureds.
- For isolation/purification step 2: filtration, you assume that 1g of product needs 100 mL of water, but the amount of water is not counted in this system. Therefore, you don't actually consider anything in regards to filtration?
- For isolation/purification step 5: distillation, you assume no solvent or chemical waste is generated, only energy input; however, you said that energy inputs where eliminated from the cost analysis. Therefore, you don't actually consider anything in regards to distillation?
- Missing reaction details (such as reagents, catalysts, etc) from Figure S2. Additionally, I would also include the next step in Path A to make the closed fused 5-ring system.
- Figure S3 is never mentioned/referenced in the text.
- Include ferrocene on Figure S4

Points of Further Discussion/Improvements:

- Why is one of the blends annealed at 110C and the other at 120C?
- The authors might want to comment on the differences in band gap shown in the optical and electrochemical measurements. In the case of the CV measurement, the MO-IDIC-2F has the largest bandgap (1.87 eV), but in the UV-Vis, MO-IDIC-2F has the higher absorption onset (i.e. smallest bandgap)
- I would recommend that the authors include the ratio of electron and hole mobilities where discussing SCLC. While high mobility is important, a balanced (i.e. similar value for both electron and holes) is important in the blend.
- The authors claim that the amberlyst15 can be recycled and reused making it better for low cost. I think that it would be valuable to include the different catalysts and the cost difference between them. Additionally, amberlyst15 isn't shown on the reaction schemes.
- In the last paragraph on Page 3, I think it is important to note a few things. IDIC and ITIC have both been published with alkyl and alkyl-phenyl. Additionally, ITIC outperforms IDIC.
- The authors should include the values in real space for the GIWAXS signals. For example, what is the distance of the π - π stacking for the neat films. [1.82 Å⁻¹ ~ 3.45 Å]
- As the authors comment on the intensity of the GIWAXS signals between the as-cast and annealed films, I would recommend that the authors have the same scaling on Figure 5c and 5f so this is clearer to the readers.
- The domain sizes are slightly larger than many high performance blends.
- Table S4 should be reformatted to make it more readable. Break up into sections for each material. This will allow you to include the titles (materials, reagents and solvents, cost, etc.) on top of each.

Reviewer #3 (Remarks to the Author):

The paper reports a new synthetic method to a key intermediate for many interesting non-fullerene acceptors, as well as a new acceptor with promising performance. In theory, I think this is very suitable for this journal. However there are unfortunately major problems with the experimental section that impact some of the claims. For example much is made in the discussion section about the high yield of some of the steps and how this leads to low cost. The one pot ring closure is stated to afford a yield of 83%. However inspection of the experimental section (MO-IDT) – shows that the actual yield reported is well over 100% (176%). 5 mmol of starting material give 8.8 mmol (5.83 g) of product according to my calculations. This suggests that significant impurities are present – perhaps due a large excess of Grignard? However, the amount of Grignard used is not stated. There are many other inconsistencies in the experimental section (see below).

I feel that any paper describing the preparation and characterization of new materials has to provide convincing evidence of their preparation and their purity. I feel this is lacking. Given the yield above 100%, and the extra carbons in ^{13}C NMR and the strange multiplicity in the ^1H NMR this is difficult to ascertain. No actual spectra are included in the SI. These should be shown, and the issues below addressed.

Given these issues, the paper needs to be resubmitted before the rest of the paper can be assessed. Especially the yield question given the subsequent emphasis on cost/yield.

Experimental

Furthermore, there is currently an insufficient level of detail in the experimental section. Were the reactions performed under inert atmosphere? Was the solvent degassed?

For MO-IDT, how much toluene was added, how much amberlyst was added (currently states a catalytic amount)? How many equivalents of OctylMgBr? What is overnight – 9h, 12h 16h? How was work-up performed? The reaction is already in toluene, so how is it extracted into DCM and water during work-up.

For MO-IDT, the ^1H NMR states a doublet ($J = 3.9\text{H}$, 24H) at 1.09 ppm. Looking at the proposed structure, I have no idea what this can possibly correspond to. Similarly the triplet at 0.76 ppm is stated to be 20H. I do not see how that is possible from the proposed structure (I would 4 terminal - CH_3). If it is overlapping with something else, how can it be a perfect triplet?

In the ^{13}C spectra, there appear 3 signals from CDCl_3 (around 70 ppm) reported as part of the structure.

For MO-IDIC, there is a singlet at 1.11 ppm (24H) – again seems not to correlate with structure. The triplet at 0.76 ppm is now 21H. How? Similar issues for MO-IDIC-2F
For MO-IDC-2F, what are the 5 ^{13}C signals from 77.32 to 69.74? These seem suspiciously like CDCl_3 . ^{19}F NMR should be reported.

Response to Reviewer 1:

1. Although the authors presented a new synthesis route to the IDIC core, the synthesis procedure for the key intermediate **4** and **4'** are not clear enough. For example, the amount of the reactant, solvent and catalyst were not given. Besides, the authors should also provide the NMR and high-resolution mass spectra of **4** and **4'** to verify the structure.

Response: We appreciate the reviewer's valuable comments. The detailed synthesis

procedure for the intermediate **4** and **4'** were added, meanwhile, the NMR and HRMS data were provided in the "Method" part on page 20, as shown below:

"Synthesis of Compound **4 and **4'**:"** Compound **3** (1.93 g 5 mmol) or **3'** (2.23 g 5 mmol) was dissolved in dry THF (60 mL) and placed under a nitrogen atmosphere. The solution was cooled to 0 °C and stirred while 32 mL hexylmagnesium bromide (0.8 M) was added dropwise. The mixture was warmed to room temperature and stirred for 12 h. It was then poured into water and extracted with dichloromethane. The organic extracts were combined and dried over anhydrous MgSO₄. After removal of the solvent, the residue was purified by silica gel chromatography (1:1, Hexanes: DCM) to give white solid **4** or **4'**.

4 (2.71g, 85%) ¹H NMR (300 MHz, CDCl₃) δ 7.39 (s, 4H), 7.23 (d, *J* = 5.3 Hz, 2H), 6.98 (d, *J* = 5.2 Hz, 2H), 1.70 (dd, *J* = 17.2, 13.9 Hz, 8H), 1.22 (s, 32H), 0.85 (t, *J* = 6.4 Hz, 12H). ¹³C NMR (75 MHz, CDCl₃) δ 143.15, 136.43, 135.27, 130.00, 128.08, 128.03, 123.65, 43.37, 31.84, 29.63, 23.67, 22.66, 14.10. HRMS (TOF) *m/z* calcd for [M+Na]⁺ C₄₀H₆₂NaO₂S₂ 661.4089, found 661.4078.

4' (3.07g, 88%) ¹H NMR (300 MHz, CDCl₃) δ 7.26 (d, *J* = 2.5 Hz, 2H), 6.94 (d, *J* = 5.3 Hz, 2H), 6.83 (s, 2H), 3.72 (s, 6H), 2.87 (s, 2H), 1.85-1.57 (m, 8H), 1.22 (s, 32H), 0.85 (t, *J* = 6.3 Hz, 12H). ¹³C NMR (75 MHz, CDCl₃) δ 149.66, 143.60, 131.79, 126.87, 124.57, 123.43, 114.93, 55.29, 41.99, 31.15, 28.97, 22.92, 21.85, 13.26. HRMS (TOF) *m/z* calcd for [M]⁺ C₄₂H₆₆O₄S₂ 698.4403, found 698.4396."

2. An interesting point is that most of the ring-closing conditions except amberlyst15 failed to yield the desirable product (especially, the chemical structure of TsOH is very similar to amberlyst15). Do the authors have any explanation on this? Besides, the source and pre-treatment of the catalyst was not included in the supporting information.

Response: Comparing with other ring-closing catalysts, amberlyst15 is a macro reticular polymeric resin with cross linked styrene divinyl benzene co-polymer which not only has high H⁺ ion exchange capacity but also has large surface area (42 m²/g), these would be the reason for the high selectivity of amberlyst15. Thus we added a sentence of “The high selectivity of amberlyst15 for the ring-closure reaction may be ascribed to the high H⁺ ion exchange capacity and large surface area of this catalyst.” to explain the reason. (page 6). The source and recycled method of amberlyst15 were added in the specific synthetic step of MO-IDT (p. 21). In addition, amberlyst15 do not need pre-treatment for this alkylated cyclization reaction.

3. The alkoxy groups on the core are usually regarded as strong electron-donating units, which should induce stronger ICT and thus result in red shift of the absorption (Energy Environ. Sci., 2017, 10, 1610-1620). The authors should explain why in this case the absorption of MO-IDIC are almost the same as IDIC.

Response: Actually, there is some red-shift of the absorption spectra of MO-IDIC in comparison with that of IDIC (see Figure 2(b)). We added a sentence to discuss and explain the phenomenon in p. 9: “In comparison with IDIC, the absorption spectrum of MO-IDIC is red-shifted slightly (red-shifted by *ca.* 10 nm for MO-IDIC film), probably due to the electron-donating effect of the methoxy substituents in MO-IDIC⁶⁷.”

4. Detailed quantitative cost calculations on a variety of OPV materials was done by the authors to evaluate the industrial potential of different materials. However, we believe that stability issue is another problem that limited the commercialization of organic solar cells. In order to further improve the quality of this manuscript, the authors should provide the stability data of the devices.

Response: In this manuscript, we mainly report our simplified synthetic route for the IDIC-like acceptors for reducing its cost and improving its photovoltaic performance. For the device stability, the measurement is under way and a long time is needed for getting the results. We reported thermal stability of the two acceptors on p. 8: “And the two acceptors possess good thermally stability up to 340 and 337°C respectively with 5% weight loss under nitrogen atmosphere, as measured by thermogravimetric analysis (TGA, Supplementary Fig. 3), which is good enough for the application in PSCs from the thermal stability point of view. It should be mentioned that the stability of the two acceptors in PSC devices should be similar with that of IDIC.”

5. The authors should include several important works in the development of highly efficient organic solar cells like: Nature Energy, (2016), 1, 15027, Nature Energy, (2016), 1, 16089, and Nature Energy, (2018), 3, 720-731, DOI: 10.1038/s41560-018-0181-5.

Response: We added the citation of the references in Ref[20]. [32] and [39]:

20. Zhao, J. *et al.* Efficient organic solar cells processed from hydrocarbon solvents.

Nat. Energy **1**, 15027 (2016).

32. Zhang, J. *et al.* Material insights and challenges for non-fullerene organic solar cells based on small molecular acceptors. *Nat. Energy* **3**,720-731 (2018).

39. Liu, J. *et al.* Fast charge separation in a non-fullerene organic solar cell with a small driving force. *Nat. Energy* **1**, 16089 (2016).

Response to Reviewer 2:

Major Comments:

1. Partial Cost Analysis: Throughout the text, the authors claim that their new synthetic process has lower energy consumption, but this is only conjecture as they provide no evidence to support their claim. As the authors included a partial cost analysis section, it is straightforward to complete this analysis fully and include energy use throughout (I also think there are other components such as hazardous waste production differences that would strengthen their claims). This is even more important as the authors make claims about energy consumption throughout the manuscript. There are a few earlier papers on this cost-analysis topic, which can help the authors strengthen this part.

Response: Compared with the synthesis of other materials with harsh reaction conditions and complex purification process, the simpler synthetic route and convenient purification process of MO-IDIC and MO-IDIC-2F are more conducive to reduce the hazardous waste production and energy consumption in the synthesis. In our calculation model, the total material costs include all the raw materials, solvent and reagent used for the reaction or purification. However, in order to simplify the calculation, the energy consumption in the cost analysis was not considered. Thus, we revised the discussions in p.7 with “**which decreases hazardous waste production**”, “**which reduces the difficulty of preparing materials and makes them more convenient to get**” and “**the synthesis of MO-IDT is more efficient and convenient**” respectively.

2. Inclusion of Extra Information: The authors included a rather large section on femtosecond-resolved transient absorption studies. I am not questioning the value which TA measurements can offer, but for this manuscript, TA doesn't really fit in. Specifically, what do we learn from these TA results? The information learned from TA doesn't support the claims that the authors are making. Because there is no comparison to materials made without their synthesis, or information learned about low cost, this information is superfluous and distracts for the themes established. The novelty of this work should focus on the synthesis and low cost attributes; therefore, I would remove this section all together.

Response: The transient absorption (TA) measurements were used for understanding why the IPCE values of the MO-IDIC-2F-based devices are much higher than that

of the MO-IDIC-based devices. Following the reviewer's comments and suggestions, we moved the TA measurement results to Supporting Information. Meanwhile, we added a sentence indicating the TA results in p. 12: “**which could be ascribed to the broader absorption in the long-wavelength range, higher electron mobility of MO-IDIC-2F and the suppressed geminate recombination in the PTQ10/MO-IDIC-2F-based devices from the transient absorption results (see the section of “Transient absorption studies” in Supplementary Information, pp. S4~S6.)**”.

3. Comparing both Yield and Performance: The authors claim that the inclusion of the methoxy helps reaction yields, but I think it is also important to compare the effects on device performance. It would be very valuable to compare IDIC and IDIC-2F throughout the text, especially in Figure 3. For example, if the methoxy group increases the yield but the device performance decreases, does the increase in yield provide more value than the decrease in performance, OR, does the device performance and the yield both increase. I think that this comparison is much more important than as cast vs annealed.

Response: We are very grateful for this valuable advice. Following the reviewer's suggestion, we compared photovoltaic performance of the acceptors of MO-IDIC-2F and IDIC-2F with PTQ10 as donor, for investigating the effect of methoxy substituents on the photovoltaic performance. Thus, we added the Supplementary Figure 5 and Supplementary Table 6, which show that the devices based on MO-IDIC-2F with methoxy substituents exhibit higher performance than that of IDIC-2F without the methoxy substituents. This indicates that methoxy substitution in MO-IDIC-2F would increase both the yield and device performance. We added a sentence on p. 11: “**In addition, the devices based on PTQ10/MO-IDIC-2F showed the higher PCE of 13.46% than that (12.09%) of the PSC based on PTQ10/IDIC-2F (see Supplementary Table 6 and Figure 5), which indicates that the methoxy substitution is beneficial to improving its photovoltaic performance.**”.

4. Materials included in Cost Analysis: In the cost analysis section, the authors include P3HT, but in Table 2 of the main text, there is no blends containing P3HT. Additionally, there have been multiple works which have yielded PCEs greater than 14%. I think that it would be important to include both of these into your cost analysis to provide a more complete understanding. As P3HT is the standard of commercial products at the moment, therefore it is very important make this comparison.

Response: We added three polymer donors of PDTB-EF-T(P2), PBDB-T-2Cl, PBDB-T-2F with PCEs great than 14%^[70-73] in the Cost analysis. Their synthetic routes were added in Supplementary Figure 23-25 and the calculated synthesis costs were added in Supplementary Table 8. The corresponding results were also added in Table 2 and Fig. 6 in main text (pp. 17-18). For P3HT, we added the comparison in Table 2 and Fig. 6 in main text, and the sentence was added on p.19:

“For the P3HT- based PSCs with O-IDTBR as acceptor, the relatively low PCE (7%)⁷⁴ results in higher C_w value (0.536 ¥/W_p).”

5. Erroneous Reaction Mechanism: The reaction mechanisms shown in Figure S1 are fundamentally incorrect, for example, the arrows show things such as carbocations attacking.

Response: Thanks for pointing out the error! The incorrect part in Supplementary Figure 1 has been corrected and modified.

Missing Data/Evidence:

1. Statistics: Why are there only statistics (average and standard deviations) for half of the blends in Table 1? Please include statistics for all blends (including SI tables). Please also include the notation for Table 1, maximum value, average value, and standard deviation.

Response: The statistics for all blends in Table 1 have been added, the maximum value and average value have been tagged. In order to ensure the reliability of the data, the data for blends in SI tables all used the average values.

2. Optimization of Reaction Conditions: The authors claim that they explored different equivalences of BF₃OEt₂, reaction time, as well as a variety of different acid catalysts like H₂SO₄, TsOH, BBr₃, AlCl₃, *etc.* For each of these, they didn't see any improvements in the yield. Please provide evidence of this. In the SI, there should be tables highlighting the different conditions and resulting yields.

Response: The results of the optimization processes for Friedel-Crafts alkylation ring-closure reaction conditions were added in Supplementary Table 1, and the related sentences on page 6 were revised as “.....Under all tested reaction conditions (variation of the number of equivalents of BF₃·OEt₂ and the reaction time), formation of these alkene molecules remained dominant (see Supplementary Table 1). In order to realize the cyclization reaction, we tried other acid catalysts like H₂SO₄, TsOH, BBr₃, AlCl₃ and so on. However, these efforts did not lead to any noticeable improvement and cannot effectively get target product (see Supplementary Table 1).”.

3. Optimization of Device Performance: The authors claim that the devices were optimized for ratio, solvents, etc. Please provide evidence highlighting the various conditions as tables in the SI.

Response: We added the photovoltaic performance parameters of the PSCs based on PTQ10: acceptors with different D:A weight ratios and different thermal annealing conditions in Supplementary Table 4 and 5. Meanwhile, the sentence on page 10: “The donor/acceptor (D/A) weight ratio in the active layer of the devices was optimized to be 1:1” was revised as “The donor/acceptor (D/A) weight ratio in the active layer of the devices and the thermal annealing temperatures were optimized, and the optimized conditions are D/A weight ratio of 1:1 and thermal annealing at

120°C (PTQ10/MO-IDIC-based PSC) and 110°C (PTQ10/MO-IDIC-2F-based PSC) for 5 min (see Supplementary Table 4 and 5)".

Minor Comments:

1. Figure 2 has the incorrect name for MO-IDIC-2F.

Response: Thanks for pointing out the error! The incorrect name for MO-IDIC-2F in Fig. 2(a) has been corrected.

2. Additionally in Figure 2, the method for energy level determination (CV/electrochem) should be mentioned in the figure caption.

Response: In the caption of Figure 2, "(the energy levels were measured by electrochemical cyclic voltammetry)" was added.

3. The authors claim that their materials exhibit strong optical absorption from 550-700 nm. I think that this number should be refined to 600-700 nm, as there is very little absorbance at 550 nm.

Response: The description of the strong optical absorption region has been revised as "from 600 to 700 nm in the solutions" on page 9.

4. The figure caption for Figure 3 doesn't include the blend or conditions for Fig 3c and 3d, i.e. which electron acceptor and is it as-cast or annealed?

Response: In the figure caption of Fig.3c and 3d, we added the device information of "based on PTQ10: MO-IDIC-2F with thermal annealing at 110 °C for 5 min".

5. There are a few sections which have incorrect grammar which needs to be addressed. I will include a few here, but this isn't intended to be a complete list. Page 1, line 4 of abstract. First sentence of introduction. First sentence on Page 5 is an incomplete sentence. There are multiple locations where the references come before the punctuation. etc.

Response: The sentence in page 1, line 4 has been revised as "while realizing the application requires further improving the PCE, increasing stability and decreasing the cost of related materials and devices." The first sentence of introduction was revised as "Solar cell, which transforms the inexhaustible solar energy into electricity, is one of the most promising clean and renewable energy sources." The first sentence on page 5 was modified as "In order to decrease the cost of IDIC and optimize the synthetic routes of *n*-OS acceptors with alkyl side chains, we firstly tried to optimize the synthetic route of Compound 2". Meanwhile, the other incorrect grammars were modified.

6. The quality of the equations and volume fractions for RSoXS discussion (page 17) are very low. Please replace with better text.

Response: The equations and volume fractions for RSoXS discussion on page 15 have been replaced with better quality equations and volume fractions.

7. The authors claim that PTQ10:MO-IDIC-2F has a fiber like structure, but looking at Figure S7b, I disagree. I do not see a fiber-like structure from the TEM.

Response: On page 15, the sentence “fiber-like structure with smaller domains was formed” was revised as “**smaller domains and interpenetrating networks were formed**”.

8. Discussion on Page 20 should be Conclusion

Response: According to the format request of Nature Communications, the Conclusion part is named as “Discussion”

9. Page 26: the authors claim their model is based on published small-scale synthetic procedures, but they do not provide references/citations for these procedures.

Response: The detailed synthetic route and processes for all the materials are shown in Supplementary Fig. 10~26, the references for these procedures were listed in Supplementary References and Supplementary Table 5. Meanwhile, we added “**(Supplementary Table 8)**” on p. 16 and p. 25 for more clear expression.

10. For isolation/purification step 2: filtration, you assume that 1g of product needs 100 mL of water, but the amount of water is not counted in this system. Therefore, you don't actually consider anything in regards to filtration?

Response: The price of water is much cheaper than that of solvents and reagents. In order to simplify the model, we do not take water consumption into account. We added a sentence in p. 27 to explain it: “**However, in considering the low cost of water in comparison with the organic solvent and reaction reagents, the amount of water is not counted in the cost calculation for simplifying the model.**”

11. For isolation/purification step 5: distillation, you assume no solvent or chemical waste is generated, only energy input; however, you said that energy inputs were eliminated from the cost analysis. Therefore, you don't actually consider anything in regards to distillation?

Response: We added a sentence in the distillation step to clearly explain the calculation model on p.27: “**We neglected the energy input in the cost calculation for simplifying the model. Actually, the distillation was used only in the first step of preparing PBDB-T and PBDB-T-2Cl, which have little impact on this statistical calculation.**”

12. Missing reaction details (such as reagents, catalysts, etc) from Figure S2. Additionally, I would also include the next step in Path A to make the closed fused 5-ring system.

Response: The corresponding reagents, catalysts and solvents are added in the Supplementary Figure 2. The synthetic route is designed to increase the yield of the corresponding compounds, however in considering the relative lower yield of Path A, we didn't do the follow-up study on this route.

13. Figure S3 is never mentioned/referenced in the text.

Response: Figure S3 is the TGA plots of MO-IDIC and MO-IDIC-2F, which is mentioned on page 8: “And the two acceptors possess good thermal stability up to 340 and 337°C respectively with 5% weight loss under nitrogen atmosphere, as measured by thermogravimetric analysis (TGA, Supplementary Fig. 3), which is good enough for the application in PSCs from the thermal stability point of view.”

14. Include ferrocene on Figure S4

Response: We added the cyclic voltammogram of ferrocene/ferrocenium (Fc/Fc^+) couple in Supplementary Figure 4, and the sentence of “the inset shows the cyclic voltammogram of ferrocene/ ferrocenium (Fc/Fc^+) couple used as an internal reference” was added in the figure caption of Supplementary Figure 4.

Points of Further Discussion/Improvements:

1. Why is one of the blends annealed at 110°C and the other at 120°C?

Response: We optimized the thermal annealing temperature for the PSCs, and found that it is 110°C for the MO-IDIC-2F-based devices and 120°C for the MO-IDIC-based devices. We added two sentences to mention the optimization results: “..... and the thermal annealing temperatures were optimized, and the optimized conditions are and thermal annealing at 120°C (PTQ10/MO-IDIC-based PSC) and 110°C (PTQ10/MO-IDIC-2F-based PSC) for 5 min (see Supplementary Table 4 and 5).” (p. 10)

2. The authors might want to comment on the differences in band gap shown in the optical and electrochemical measurements. In the case of the CV measurement, the MO-IDIC-2F has the largest bandgap (1.87 eV), but in the UV-Vis, MO-IDIC-2F has the higher absorption onset (i.e. smallest bandgap)

Response: The bandgap from electrochemical measurement (E_g^{ec}) is usually larger than that from optical measurement (E_g^{opt}), and the difference between E_g^{ec} and E_g^{opt} is dependent on the molecular structure of the conjugated molecules. The reason could be explained that there exist over potentials for the oxidation and reduction of the conjugated molecules (charge transfer between the electrode and conjugated molecules), which can enlarge the E_g^{ec} . In addition, the overpotential value is related to the molecular structure and the interfacial structure of the molecules film deposited on the electrode. Actually, the energy level obtained from the electrochemical measurement should be more reasonable in the consideration of charge transfer in the active layers of PSCs, because the influence of the molecular structure on the charge transfer is included in the energy level values measured by electrochemical method.

3. I would recommend that the authors include the ratio of electron and hole mobilities where discussing SCLC. While high mobility is important, a balanced (i.e. similar value for both electron and holes) is important in the blend.

Response: The electron and hole mobilities of the blend films and the ratio between them were added in the Supplementary Table 3.

4. The authors claim that the amberlyst15 can be recycled and reused making it better for low cost. I think that it would be valuable to include the different catalysts and the cost difference between them. Additionally, amberlyst15 isn't shown on the reaction schemes.

Response: Amberlyst15 is currently the most suitable catalyst for this reaction with high yield and high selectivity. Other catalysts show poor selectivity and low yield, which can't be used in the synthetic route. So we didn't calculate their cost difference. In addition, the amberlyst15 as catalyst has been added in the reaction scheme in Fig. 1.

5. In the last paragraph on Page 3, I think it is important to note a few things. IDIC and ITIC have both been published with alkyl and alkyl-phenyl. Additionally, ITIC outperforms IDIC.

Response: As the reviewer pointed out, the core of IDIC and ITIC have both been published with alkyl and alkyl-phenyl. In terms of low cost preparation, the advantage of IDIC lies in its simpler structure. Thus, the sentence on page 3: "IDIC with the alkyl side chains on IDT core, possesses the advantage of simple alkyl side chains in comparison with the alkyl-phenyl side chains in ITIC. In addition, the smaller fused-ring in its central unit makes IDIC have the potential to be the low cost acceptor." was revised as "IDIC with the alkyl side chains on IDT core, possesses the advantage of the smaller fused-ring in its central unit in comparison with ITIC, which makes IDIC have the potential to be the low cost acceptor." In addition, the photovoltaic performance of IDIC and ITIC depends on the donor materials used in the devices. For the small molecule donors and some polymers like PTQ10, IDIC exhibits better photovoltaic performance than ITIC.

6. The authors should include the values in real space for the GIWAXS signals. For example, what is the distance of the π - π stacking for the neat films. [1.82 Å⁻¹ ~ 3.45 Å]

Response: The distance of the π - π stacking for the neat films were added with the sentences: "(the π - π stacking distance was 3.55 Å)" and "The π - π stacking distance of these molecules are 3.41 Å for MO-IDIC-2F and 3.45 Å for MO-IDIC". (p. 13)

7. As the authors comment on the intensity of the GIWAXS signals between the as-cast and annealed films, I would recommend that the authors have the same scaling on Figure 5c and 5f so this is clearer to the readers.

Response: The scaling in Fig. 5e and Fig. 5f has been modified to the same scaling.

8. The domain sizes are slightly larger than many high performance blends.

Response: Yes, the domain sizes are slight larger, but they are similar with some high PCE systems with the domain size from 36 to 42 nm: 1. Fan, Q. et al. *Sci China*

Chem. 61. 531-537 (2018). 2. Luo, Z. et al. *Adv. Mater.* 30, 1706124 (2018). 3. Zhang, Z. et al. *Adv. Funct. Mater.* 28, 1705095 (2018).

9. Table S4 should be reformatted to make it more readable. Break up into sections for each material. This will allow you to include the titles (materials, reagents and solvents, cost, etc.) on top of each.

Response: Supplementary Table 8 (original Supplementary Table 4) was modified according to the reviewer's suggestion. The compounds used in the synthetic route of several materials were divided into reagents, solvent and purification for more readable.

Response to Reviewer 3:

I feel that any paper describing the preparation and characterization of new materials has to provide convincing evidence of their preparation and their purity. I feel this is lacking. Given the yield above 100%, and the extra carbons in ^{13}C NMR and the strange multiplicity in the ^1H NMR this is difficult to ascertain. No actual spectra are included in the SI. These should be shown, and the issues below addressed. Given these issues, the paper needs to be resubmitted before the rest of the paper can be assessed. Especially the yield question given the subsequent emphasis on cost/yield.

Response: In the "Method" part, the detailed synthesis procedures, NMR and HRMS for the intermediate molecules **4** and **4'** were added. For the synthesis of compound MO-IDT, we corrected the errors in the yield data. In order to show clearly the purity of compounds, ^1H NMR and ^{13}C NMR spectra of all new materials were added. We added a sentence on p. 7 to indicate the measurement results: "The structure and purity of the compounds were measured and confirmed by ^1H NMR and ^{13}C NMR spectra, as shown in Supplementary Figures 27 ~ 36." And a sentence on p.8: "The chemical structures of the two *n*-OS acceptors were characterized by ^1H and ^{13}C NMR (Supplementary Figures 37~40)."

Experimental

1. Furthermore, there is currently an insufficient level of detail in the experimental section. Were the reactions performed under inert atmosphere? Was the solvent degassed?

Response: We supplemented some reaction details in the "Method" section, including adding a sentence "The solvents do not need to degas in all the reactions, and the reactions all performed under a nitrogen atmosphere." in the section of "Materials and synthesis", and adding "under a nitrogen atmosphere" and "8 mL NaOH aqueous solution (2M) was added" in the section of "Synthesis of Compound **3'**".

2. For MO-IDT, how much toluene was added, how much amberlyst was added (currently states a catalytic amount)? How many equivalents of OctylMgBr? What

is overnight -9h, 12h 16h? How was work-up performed? The reaction is already in toluene, so how is it extracted into DCM and water during work-up.

Response: We added a section of “**Synthesis of Compound MO-IDT**” (p. 21) in “**Method**” part to describe the detailed synthetic processes of **MO-IDT**: “**Synthesis of Compound MO-IDT:** Compound **4**’ (2.09 g, 3 mmol) was dissolved in dry toluene (30 mL) and placed under a nitrogen atmosphere, then amberlyst15 (2 g) as catalytic (Acros Amberlyst15, (dry) ion-exchange resin) was added and heated at 85 °C for 12h. After the reaction, the mixture was filtered and the organic liquids were collected. The catalytic was washed with dichloromethane for recycle (the catalytic was cleaned in distilled water, dilute hydrochloric acid, distilled water, and ethanol in sequence, then drying). After removal of the solvent under reduced pressure, the residue was purified by column chromatography on silica gel using petroleum ether as eluent to give yellow solid MO-IDT (1.83 g, 92% yield).”

3. For MO-IDT, the ¹H NMR states a doublet (J = 3.9H, 24H) at 1.09 ppm. Looking at the proposed structure, I have no idea what this can possibly correspond to. Similarly the triplet at 0.76 ppm is stated to be 20H. I do not see how that is possible from the proposed structure (I Would 4 terminal -CH₃). If it is overlapping with something else, how can it be a perfect triplet? For MO-IDIC, there is a singlet at 1.11 ppm (24H) again seems not to correlate with structure. The triplet at 0.76 ppm is now 21H. How? Similar issues for MO-IDIC-2F.

Response: We added the ¹H NMR spectra of the corresponding molecules in Supplementary Figures 27, 29, 31, 33, 35, 37, 39. From the ¹H NMR spectra, the compounds of **4** and **4**’ show the perfect triplet signals at 0.85 ppm (12H), which belong to the terminal -CH₃. After the cyclization reaction, the triplet signals around 0.7 ppm show 20H, which would be the overlap of the terminal -CH₃ and adjacent hydrogen atoms signals (-CH₂-). Thus the data around 0.7 ppm for compounds MO-IDT, **7**, MO-IDIC and MO-IDIC-2F were replaced by “0.72-0.80 (m, 20H)”, “0.72-0.81 (m, 20H)”, “0.72-0.81 (m, 20H)”, “0.73-0.80 (m, 20H)” respectively. In the same cases, the ¹H NMR of MO-IDT at 1.09 ppm was modified to “1.06-1.12 (m, 24H)” and the data of MO-IDIC-2F at 1.11 ppm was modified to “1.08-1.16 (m, 24H)”.

4. In the ¹³C spectra, there appear 3 signals from CDCl₃ (around 70 ppm) reported as part of the structure. For MO-IDC-2F, what are the 5 ¹³C signals from 77.32 to 69.74? These seem suspiciously like CDCl₃.

Response: In the data of ¹³C NMR for all new materials, the signals around 70 ppm are really belong to CDCl₃, which were deleted. Meanwhile, we added the ¹³C NMR spectra of the corresponding compound in Supplementary Figures 28, 30, 32, 34, 36, 38, 40.

5. ¹⁹F NMR should be reported.

Response: The ^{19}F NMR spectrum of MO-IDIC-2F was added in the Supplementary Figure 41. And a sentence of “The chemical structures of the two *n*-OS acceptors were characterized by ^1H and ^{13}C NMR (Supplementary Figures 37~40), ^{19}F NMR spectrum (Supplementary Figure 41 for MO-IDIC-2F)” was added on p. 8.

Reviewers' comments:

Reviewer #1 (Remarks to the Author):

The paper can be accepted now

Reviewer #2 (Remarks to the Author):

Summary:

In this manuscript, Li et al. synthesized a new electron donating fused ring core, called MO-IDT, which is the previously published C6-IDIC core with a new methoxy substituent on the core benzene unit. They paired this new core with both INCN and mono fluorinated INCN acceptor pendants to make two fused-ring electron acceptors of MO-IDIC and MO-IDIC-2F. These acceptors yielded PCEs between 10-13% when paired with a previously reported PTQ-10 polymer donor. The authors highlight two major features of this material. First, the synthesis of the donor core has been redesigned to offer a higher yield, and second, because of this design, the PTQ-10:Acceptor blends offer a lower cost compared to other blends. The authors made appropriate changes to the manuscript based on the feedback from reviews. After addressing the below issues, the work should be ready for publication.

Comments:

- This manuscript seems to be rather large/long for a communication. Is the total word count in the range acceptable for Nature Communications?
- The following figures have very low quality – please fix: Figure 2, Figure 3, Figure 6
- There is no Figure 4 in the manuscript.
- The authors need to provide references/citations which illustrate that their cost analysis is appropriate/matches literature precedent. Please include appropriate citations on page 16, line 397 and/or page 25, line 649 (i.e. cite other papers which do a cost analysis). This will help validate the cost analysis.
- Consider redrawing the molecular structures in Figure 2 with the correct stereochemistry about the double bond between donor and acceptor moieties. There have been multiple publications that have shown the oxygen of the INCN acceptor and sulfur of the last thiophene are closest, i.e. you need to flip entire acceptor moiety. See either of these for correct structure: DOI: 10.1039/c7tc01310h or DOI: 10.1002/adma.201700254. You can also confirm this is the structure of your materials by either (a) growing the single crystal or (b) looking at 2D NMR signals between the thiophene and on the double bond Hs.

Reviewer #3 (Remarks to the Author):

The authors have implemented some changes, but I still find inaccuracies throughout the manuscript, which are frustrating and not of the standard I would expect for this journal. Some of them are important (as detailed below). I also find the cost analysis not to be very information. The difference between the synthesis of a first discovery molecule, and the actual synthesis (once optimized) of almost any chemical entity (for example a medicinal drug) is completely different. None of the suggested synthetic routes would be scaled-up as first published. There are ample opportunities to optimize all of them. For example chromatography (as currently used in the synthesis of 4) is not scalable. In addition heating times have been ignored, as have cooling. In the synthesis of their MO-IDIC acceptor, the preparation of the starting material 4 needs a low temperature (-78C) lithiation step for example. This is incredibly expensive on a large scale (much more than heating). I think the

authors have to make it very clear that this is a very crude estimate.

Given the important the authors give to yield and scalability, it is important that the authors comment how reproducible the synthetic steps are. Are these average yields? How many repeats etc.

Further specific points:

- In the text, the authors highlight the improved synthesis of route b to make 2, versus route a. However their analysis starts with compounds 1 and 3 respectively. Given that the synthesis of 3 is more difficult and expensive than 1, I'm not sure that this is reasonable.
- Similarly line 166/167: the author's state; 'Only three steps in the synthesis of MO-IDT are much simpler than that in preparing the core of IDIC (fig 1a)'. Again this is bending the truth in my opinion. The synthesis of IDIC in figure 1a is given from the cheapest starting material (dibromoxylene). However for MO-IDT, the synthesis starts from an advanced intermediate (compound 5). How is compound 5 made? The authors state (line 158) Compound 5 (Fig. 1c) was synthesized according to the method in the literature (ref 65). However compound 5 is not reported in reference 65 (only the dioctyl version, not the dimethyl). Assuming the method is the same, then the synthesis of 5 requires 3 steps itself according to ref 65. The total synthesis route is 6 steps therefore, not 3 as claimed. Therefore the conclusions that follow regarding cost and number of steps are somewhat suspect and should be rewritten accordingly.
- The ¹H NMR data is still incorrect! For compound 4 and 4', what can the signal specified as a singlet at 1.22 ppm (32H) be? Surely this is a broad signal. Same problem for compound 7 and MO-IDIC. Compound 4' has two -OH ¹H signals in the ¹H NMR. 4 does not. Why.
- Line 210; new acceptors show 'strong optical absorption'. Please quantify this statement – what are ext. coefficients.
- In the synthesis of 3, the catalyst amounts are mixed up. The ligand is 144 mg which is 0.5 mmol (10 mol%) to starting material. However only 0.56 mg Pd(acetate) is used. This is claimed to 0.25 mmol (5% mol%), but is actually 0.0025 mmol! Which is correct? In the cost calculation table (supplementary table 8) which mol% has been used? This can make a big impact. Currently it is not easy to see this information.
- In the cost analysis, (supplementary table 8). The authors give the cost the 3-Thenoic Acid. However they do not use this in the synthesis (they use Ethyl-3-thiophene carboxylate), which is much more expensive. They do not include the preparation of this ester from the acid in their cost analysis (no ethanol used, no sulfuric acid). Given these esterification steps are used in other cases, (eg ITIC); why is it not included?
- Experimental info. Line 484, states: Compound 5, 484 6 and PTQ10 were synthesized according to the procedures reported in the literatures⁴⁶. Reference 46 is a review on P3HT and does not contain this info
- Line 590. What is PDINO? Reference? Where was it purchased from?
- The synthesis procedure for 2 (From 4) is not reported anywhere.

Scheme 1: Catalyst not included in the synthesis of 3'

Fig 5: Please states that the line cuts are offset for clarity. The intensity axis should be changes

accordingly for each blend

Response to Reviewer 2

1. This manuscript seems to be rather large/long for a communication. Is the total word count in the range acceptable for Nature Communications?

Response: The number of words in this manuscript meet the requirements of Nature Communications.

2. The following figures have very low quality please fix: Figure 2, Figure 3, Figure 6.

Response: Fig.2, Fig.3 and Fig.5 (original Fig.6) have been replaced with better quality figures.

3. There is no Figure 4 in the manuscript.

Response: The error has been corrected in the revised manuscript.

4. The authors need to provide references/citations which illustrate that their cost analysis is appropriate/matches literature precedent. Please include appropriate citations on page 16, line 397 and/or page 25, line 649 (i.e. cite other papers which do a cost analysis). This will help validate the cost analysis.

Response: we revised the related sentence (on page 16, lines 4~6) into: “we built a model (see the section of “methods”) and present a detailed quantitative cost calculations based on the synthetic procedures and evaluation rules published in literatures⁶⁹⁻⁷¹”. The citation of the references in Ref [69], [70], [71] were added.

69. Po, R & Roncali, J. Beyond efficiency: scalability of molecular donor materials for organic photovoltaics. *J. Mater. Chem. C* **4**, 3677-3685 (2016).

70. Kalowekamo, J & Baker, E. Estimating the manufacturing cost of purely organic solar cells. *Sol. Energy* **83**, 1224-1231 (2009).

71. Xue, R. Zhang, J. Li, Y & Li, Y. F. Organic Solar Cell Materials toward Commercialization. *Small* **14**, 1801793 (2018).

5. Consider redrawing the molecular structures in Figure 2 with the correct stereochemistry about the double bond between donor and acceptor moieties. There have been multiple publications that have shown the oxygen of the INCN acceptor and

sulfur of the last thiophene are closest, i.e. you need to flip entire acceptor moiety. See either of these for correct structure: DOI: 10.1039/c7tc01310h or DOI: 10.1002/adma.201700254. You can also confirm this is the structure of your materials by either (a) growing the single crystal or (b) looking at 2D NMR signals between the thiophene and on the double bond Hs.

Response: The molecular structures in Fig.2 have been modified. In addition, we added the crystal structures of MO-IDIC and MO-IDIC-2F for confirming the molecular structures in Supplementary Figure 44 and 45. The sentence of “Moreover, the single crystals of these two molecules further clarify the structures (Supplementary Figure 44~45)” was added on p.7, the last two lines.

Response to Reviewer 3

Some of them are important (as detailed below). I also find the cost analysis not to be very information. The difference between the synthesis of a first discovery molecule, and the actual synthesis (once optimized) of almost any chemical entity (for example a medicinal drug) is completely different. None of the suggested synthetic routes would be scaled-up as first published. There are ample opportunities to optimize all of them. For example chromatography (as currently used in the synthesis of 4) is not scalable. In addition heating times have been ignored, as have cooling. In the synthesis of their MO-IDIC acceptor, the preparation of the starting material 4 needs a low temperature (-78C) lithiation step for example. This is incredibly expensive on a large scale (much more than heating). I think the authors have to make it very clear that this is a very crude estimate.

Response: In this manuscript, we mainly report our simplified synthetic route for the IDIC-like acceptors for reducing its cost and improving its photovoltaic performance. In addition, we built a model based on the published small-scale synthetic procedures and evaluation rules published in literatures. In this model, we mainly quantitatively analyze the synthetic cost of the materials and neglected the energy input in the cost calculation for simplifying the model (the statement can be seen in “Quantitative analysis of the material costs” part). Thus we added a sentence on p.16 to indicate roughness of the calculation: “However, the energy input in the cost calculation was neglected for simplifying this model. Therefore, the cost calculated in the model cannot represent the actual cost of mass production, but can roughly reflect the relative costs of active layer materials.”

1. Given the important the authors give to yield and scalability, it is important that the authors comment how reproducible the synthetic steps are. Are these average yields? How many repeats etc.

Response: The yields reported in this work are average. In addition, for the reproducibility of these synthetic steps, we and the other two laboratory staffs repeated the synthetic steps more than 10 times.

2. In the text, the authors highlight the improved synthesis of route b to make 2, versus

route a. However their analysis starts with compounds 1 and 3 respectively. Given that the synthesis of 3 is more difficult and expensive than 1, I'm not sure that this is reasonable.

Response: In order to better compare routes A and B, we added raw materials **8** and **9** in Supplementary Figure 2. The synthetic step of compound 3 is the same as 3', which can be more convenient and simple to prepare than compound 1. The synthesis process of 3 was added in the "Method" part (see p. 20). In addition, compounds **8** and **9** can be easily purchased in Innochem with 1368 ¥/100g for **8** and 2325 ¥/100g for **9**, although the price of **9** is slightly higher, the more effective synthesis step of compound **3** with higher yield leads the synthesis path B more suitable.

3. Similarly line 166/167: the author's state; 'Only three steps in the synthesis of MO-IDT are much simpler than that in preparing the core of IDIC (fig 1a)'. Again this is bending the truth in my opinion. The synthesis of IDIC in figure 1a is given from the cheapest starting material (dibromoxylene). However for MO-IDT, the synthesis starts from an advanced intermediate (compound 5). How is compound 5 made? The authors state (line 158) Compound 5 (Fig. 1c) was synthesized according to the method in the literature (ref 65). However compound 5 is not reported in reference 65 (only the dioctyl version, not the dimethyl). Assuming the method is the same, then the synthesis of 5 requires 3 steps itself according to ref 65. The total synthesis route is 6 steps therefore, not 3 as claimed. Therefore the conclusions that follow regarding cost and number of steps are somewhat suspect and should be rewritten accordingly.

Response: We are very grateful for this valuable advice. The synthesis processes for the core of IDIC and MO-IDT have been adjusted. Compounds 2,5-dibromo-terephthalic acid diethyl ester and 1,4-Dibromo-2,5-dimethoxybenzene, which are easy to purchase in large quantities, were used to prepare the core of IDIC and MO-IDT, respectively. Compound 5 can be prepared from 1,4-Dibromo-2,5-dimethoxybenzene in one step, thus the sentence in the first two lines on p. 7 was revised as "Only four steps in the synthesis of MO-IDT are simpler than that in preparing the core of IDIC with six steps". In addition, the Supplementary Table 8, 9 and Fig.5 in main text have been modified accordingly.

4. The ¹H NMR data is still incorrect! For compound 4 and 4', what can the signal specified as a singlet at 1.22 ppm (32H) be? Surely this is a broad signal. Same problem for compound 7 and MO-IDIC. Compound 4' has two -OH 1H signals in the ¹H NMR. 4 does not. Why.

Response: The single signal at about 1.2 ppm may be due to the relative low resolution and sensitivity of 300MHz NMR. So we re-tested the ¹H NMR of compound 4, 4', 7 and MO-IDIC with 400MHz NMR and modified the ¹H NMR data of these molecules. In addition, the ¹H NMR spectrums of these compounds were added in Supplementary Figures 31,33, 37, 39. For the signal of hydroxyl hydrogen in compound 4, it is theoretically affected by many factors, such as solvent,

concentration and molecular structure. And the active hydrogen signal is not fixed, so there is no obvious signal, in addition, the HRMS (TOF) of compound 4 can further confirm the correctness of the structure.

5. Line 210; new acceptors show 'strong optical absorption'. Please quantify this statement-what are ext. coefficients.

Response: The sentence: "These *n*-OS molecules exhibit strong optical absorption from 600 to 700 nm in the solutions, and MO-IDIC and MO-IDIC-2F exhibit the absorption peak at 670 and 677 nm respectively" on p. 9 has been revised as "These *n*-OS molecules exhibit strong optical absorption from 600 to 700 nm in the solutions with a maximum extinction coefficient of $2.8 \times 10^5 \text{ M}^{-1} \text{ cm}^{-1}$ at 670 nm for MO-IDIC and $2.9 \times 10^5 \text{ M}^{-1} \text{ cm}^{-1}$ at 677 nm for MO-IDIC-2F." (see p. 8~9)

6. In the synthesis of 3, the catalyst amounts are mixed up. The ligand is 144 mg which is 0.5 mmol (10 mol%) to starting material. However only 0.56 mg Pd(acetate) is used. This is claimed to 0.25 mmol (5% mol%), but is actually 0.0025 mmol! Which is correct? In the cost calculation table (supplementary table 8) which mol% has been used? This can make a big impact. Currently it is not easy to see this information.

Response: Thanks for pointing out the error! For the Pd(OAc)₂, the 0.25 mmol (5% mol%) is used and the sentence was corrected to: "Pd(OAc)₂ (56 mg, 0.25 mmol)" (see p. 20 in the Section of "Synthesis of Compound 3 and 3'"). In the cost calculation Table (supplementary Table 8), the calculated amount is consistent with the synthesis step.

7. In the cost analysis, (supplementary table 8). The authors give the cost the 3-Thenoic Acid. However they do not use this in the synthesis (they use Ethyl-3-thiophene carboxylate), which is much more expensive. They do not include the preparation of this ester from the acid in their cost analysis (no ethanol used, no sulfuric acid). Given these esterification steps are used in other cases, (eg ITIC); why is it not included?

Response: In order to further unify the standard, we revised the process of price accountingly and used esterified products to compare prices. Compounds 2,5-dibromo-terephthalic acid diethyl ester and 2-bromothiophene-3-carboxylic acid ethyl ester were used in price calculation instead of raw materials. In addition, the Supplementary Table 8, 9 and Table 2, Fig.5 in main text have been adjusted accordingly.

8. Experimental info. Line 484, states: Compound 5, 6 and PTQ10 were synthesized according to the procedures reported in the literatures⁴⁶. Reference 46 is a review on P3HT and does not contain this info

Response: The references have been revised as: "the procedures reported in the literatures^{65, 29, 49}."

9. Line 590. What is PDINO? Reference? Where was it purchased from?

Response: We add the sentence: “PDINO was synthesized in our lab according to the procedures reported in the literature⁷⁸” on p. 24, and the cited Ref [78] is:

78. Zhang, Z.-G. et al. Perylene Diimides: a Thickness-insensitive Cathode Interlayer for High Performance Polymer Solar Cells. *Energy Environ. Sci.*, **7**, 1966-1973 (2014).

10. The synthesis procedure for 2 (From 4) is not reported anywhere.

Response: The synthesis procedure for 2 was added on p. 21 in the “**Method**” part as shown below:

Synthesis of Compound 2 and MO-IDT: Compound 4 (1.98 g, 3mmol) or 4' (2.09 g, 3 mmol) was dissolved in dry toluene (30 mL) and placed under a nitrogen atmosphere, then amberlyst15 (2 g) as catalyst (Acros Amberlyst15, (dry) ion-exchange resin) was added and heated at 85 °C for 12 h. After the reaction, the mixture was filtered and the organic liquids were collected. The catalyst was washed with dichloromethane for recycle (the catalyst was cleaned in distilled water, dilute hydrochloric acid, distilled water, and ethanol in sequence, then drying). After removal of the solvent under reduced pressure, the residue was purified by column chromatography on silica gel using petroleum ether as eluent to give white solid 2 (from Compound 4) or MO-IDT (from Compound 4').

2 (1.14g, 63% yield). ¹H NMR (400 MHz, CDCl₃) δ 7.27 (s, 2H), 7.25 (d, *J* = 4.8 Hz, 2H), 6.96 (d, *J* = 4.8 Hz, 2H), 2.03-1.79 (m, 8H), 1.20-1.02 (m, 24H), 0.85-0.73 (m, 20H). ¹³C NMR (101 MHz, CDCl₃) δ 155.10, 153.23, 141.66, 135.59, 126.13, 121.72, 113.13, 53.67, 39.21, 31.62, 29.73, 24.15, 22.60, 14.05. HRMS (TOF) *m/z* calcd for [M]⁺C₄₀H₅₈S₂ 602.3980, found 602.3983.

11. Scheme 1: Catalyst not included in the synthesis of 3'. Fig 5: Please states that the line cuts are offset for clarity. The intensity axis should be changes accordingly for each blend

Response: The catalysts used in the synthesis of 3 and 3' have been added in Fig. 1. The statement for the line cuts was added on p. 25 in the part of “**GIWAXS Characterization**”: “1D profile was obtained with the intensity distribution analysed along in-plane and out-of-plane direction. It should be noted that the region at low *q* position was blocked by beamstop, and thus no signal could be observed in 1D profile at the low *q* position.”. The intensity axis have been modified in Fig. 4.

REVIEWERS' COMMENTS:

Reviewer #3 (Remarks to the Author):

The authors have correct the experimental and updated the discussion. In general I think the paper can be published.

However the new addition (line 400) 'However, the energy input in the cost calculation was neglected for simplifying this model. Therefore, the cost calculated in the model cannot represent the actual cost of mass production, but can roughly reflect the relative costs of active layer materials.'

I still think is a very unrealistic simplification. Energy input, especially for cooling, has one of the largest cost implications for any synthesis (much larger in many cases than starting material costs). I would suggest it is clarified as follows:

However the energy input, either heating or cooling, in the cost calculation was ignored to simplify the calculations. Therefore the calculations are unlikely to reflect the actual cost of mass production, but can be used as a rough indication of synthetic complexity.

COMMENTS TO AUTHOR:

Reviewer 3

The authors have corrected the experimental and updated the discussion. In general I think the paper can be published. However the new addition (line 400) 'However, the energy input in the cost calculation was neglected for simplifying this model. Therefore, the cost calculated in the model cannot represent the actual cost of mass production, but can roughly reflect the relative costs of active layer materials. I still think is a very unrealistic simplification. Energy input, especially for cooling, has one of the largest cost implications for any synthesis (much larger in many cases than starting material costs). I would suggest it is clarified as follows: However the energy input, either heating or cooling, in the cost calculation was ignored to simplify the calculations. Therefore the calculations are unlikely to reflect the actual cost of mass production, but can be used as a rough indication of synthetic complexity.

Response: We agree with the reviewer's comments and revision suggestions, and revised the related sentence as: "In this model **of the cost calculation**, all the raw materials, solvent and reagent used for the reaction or purification were taken into account in order to estimate the total material costs. **However the energy input, either heating or cooling, was ignored to simplify the calculations. Therefore the calculations are unlikely to reflect the actual cost of mass production, but can be used as a rough indication of synthetic complexity.**"